# Sensitivity to image recurrence across eye-movement-like image transitions through local serial inhibition in the retina

**Vidhyasankar Krishnamoorthy[1,2], Michael Weick[1], Tim Gollisch[1,2]\***

[1]Department of Ophthalmology, University Medical Center Göttingen, Bernstein Center for Computational Neuroscience Göttingen, Göttingen, Germany; [2]Visual Coding Group, Max Planck Institute of Neurobiology, Martinsried, Germany

**Abstract** Standard models of stimulus encoding in the retina postulate that image presentations activate neurons according to the increase of preferred contrast inside the receptive field. During natural vision, however, images do not arrive in isolation, but follow each other rapidly, separated by sudden gaze shifts. We here report that, contrary to standard models, specific ganglion cells in mouse retina are suppressed after a rapid image transition by changes in visual patterns across the transition, but respond with a distinct spike burst when the same pattern reappears. This sensitivity to image recurrence depends on opposing effects of glycinergic and GABAergic inhibition and can be explained by a circuit of local serial inhibition. Rapid image transitions thus trigger a mode of operation that differs from the processing of simpler stimuli and allows the retina to tag particular image parts or to detect transition types that lead to recurring stimulus patterns.

**\*For correspondence:** tim.
gollisch@med.uni-goettingen.de

**Competing interests:** The
authors declare that no
competing interests exist.

**Reviewing editor:** Alexander
Borst, Max Planck Institute of
Neurobiology, Germany

## Introduction

Neurons in sensory systems encode stimuli in their electrical activity. Often, this encoding is thought to be largely captured by the cells' receptive fields, so that a cell's activity reflects an increase of the preferred stimulus feature inside its receptive field. In the retina, for example, neurons generally respond strongly to local or global increases in preferred visual contrast inside their spatial receptive fields (*Kuffler, 1953*; *Enroth-Cugell and Robson, 1966*; *Meister and Berry, 1999*). Thus, when a new image comes into view, increases in spiking activity of ganglion cells are expected to represent image parts that contain large local contrast changes. During natural vision, however, images do not arrive in isolation, but are embedded in a continuous, strongly structured stream of visual stimulation. Brief periods of relatively stable fixation are interspersed with sudden shifts in the direction of gaze ('saccades'), so that images come in rapid sequences, separated by brief transition periods. Saccades of the eyes or head are ubiquitous among vertebrates (*Land, 2015*), including mice (*Sakatani and Isa, 2007*; *Keller et al., 2012*; *Wang et al., 2015*), and they can exert a strong influence on ganglion cell spiking, evoking transient spike bursts in some ganglion cells (*Noda and Adey, 1974*) and suppressing activity in others by triggering inhibition in the retinal network (*Roska and Werblin, 2003*).

Yet, despite the abundance of saccades and their strong influence on the natural flow of visual stimuli, relatively little is known about how these natural stimulus dynamics influence the processing of visual information in early vision. In particular, systematic investigations are lacking of how the previously fixated visual pattern affects the encoding of a newly arriving image. Moreover, the rapid image sequences and the strong, global motion signal during a saccadic transition appear to trigger specific excitatory and inhibitory circuit elements (*Passaglia et al., 2001*; *Roska and Werblin, 2003*;

*Geffen et al., 2007*), which may alter the processing of visual information compared to responses under simpler, more traditional laboratory stimuli.

In particular, interactions of local excitatory signals from presynaptic bipolar cells with the inter-connected network of amacrine cells have an enormous potential for shaping ganglion cell responses in a nonlinear, stimulus-context-dependent fashion (*Gollisch and Meister, 2010*; *Schwartz and Rieke, 2011*; *Taylor and Smith, 2011*; *Masland, 2012b*). Various examples have shown that such interactions can mediate specific, functionally relevant response characteristics beyond contrast detection inside the receptive field for a number of ganglion cell types (*Olveczky et al., 2003*; *Münch et al., 2009*; *Bölinger and Gollisch, 2012*; *Vaney et al., 2012*; *Zhang et al., 2012*). The examples have led to the hypothesis that many amacrine cells function in a highly task-specific manner (*Masland, 2012a*), making it difficult to study their influence on retinal signal processing with simple, generic stimuli and suggesting to systematically explore the characteristics of stimulus encoding under more natural stimulus dynamics. A particularly suited stimulus context appears to be given by eye-movement-like rapid image transitions. These are characterized by a fairly stereotypic temporal structure of alternating image fixations and shifts, yet the rapid image sequences and the strong, global motion stimulus during a transition can be expected to trigger various neuronal interactions in the retinal network.

We therefore investigated how ganglion cells in the mouse retina encode stimulus transitions across saccade-like image shifts. This revealed that, contrary to common stimulus–response models, particular ganglion cells robustly responded to recurrences of the same image across the transition, but were suppressed when the image position changed. Finally, a combination of pharmacological interventions, intracellular recordings, and computational modeling indicated that this unexpected sensitivity to image recurrence is mediated by a circuit of serial inhibition, demonstrating how saccade-triggered inhibitory interactions can alter retinal stimulus encoding.

## Results

### Sensitivity to image recurrence across saccade-like transitions

To explore how the retina encodes visual information in the presence of saccade-like image transitions, we investigated how responses of ganglion cells in the mouse retina depend on the spatial pattern of the fixated images prior to and after the transition. We recorded the spiking activity of individual ganglion cells in the isolated mouse retina with multi-electrode arrays while stimulating the photoreceptors with saccade-like shifts of a spatial grating. After each shift, the grating stopped randomly for a brief fixation at one out of four different spatial phases, here called positions 1 to 4 (*Figure 1A*), resulting in 16 possible transitions. Each transition occurred many times in this random sequence, and we collected the responses from all instances of a transition to compute a firing rate profile (*Figure 1B*, top). To compare responses from all 16 transitions, we arranged these profiles in a matrix (*Figure 1B*, bottom) to denote the dependence on the positions prior to the transition ('starting position') and afterwards ('target position'). The response matrix of *Figure 1B* shows that this particular sample cell preferred certain target positions (here position 2 or 3), and responses were further enhanced when the starting position was among the reversed positions (here 1 and 4). This response structure is consistent with the classical idea of receptive field activation, yielding the strongest responses when, after the transition, the stimulus inside the receptive field (better) matches the cell's preferred contrast.

Yet, a distinct subset of recorded ganglion cells showed strikingly different response characteristics. The sample cell of *Figure 1C* fired a spike burst during each saccade-like transition. It then displayed a second pronounced spike burst about 100 to 150 ms after the onset of the new fixation, but only for such transitions where the grating returned to the same spatial phase, that is, where starting and target position were equal. Clearly, this spike burst during the new fixation did not encode the occurrence of a particular preferred grating position or an increase in preferred contrast. Rather, this response marked such transitions across which changes in the stimulus pattern were minimal.

These distinct response peaks for transitions with equal starting and target position were characteristic for about 5–10% of the recorded ganglion cells, and we will here refer to these cells as image-recurrence-sensitive (IRS) cells (see 'Materials and methods' for a quantitative criterion of

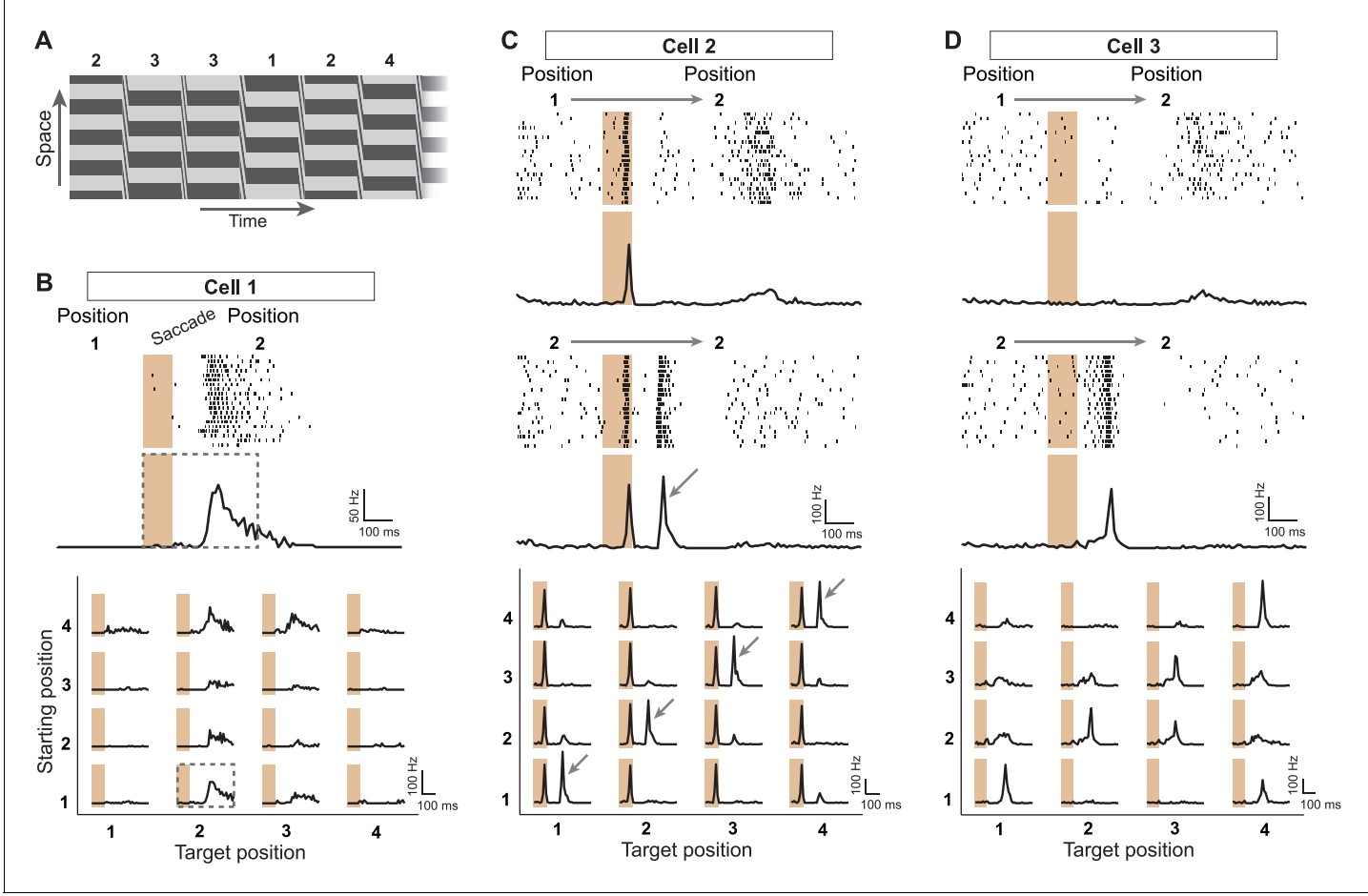

**Figure 1.** Sensitivity to image recurrence by retinal ganglion cells under saccade-like image shifts. (A) Space-time representation of the saccade stimulus. The stimulus consisted of a spatial grating that was repeatedly shifted in a saccade-like fashion during 100 ms and then stopped randomly at one out of four different grating positions, where it then remained fixed for 800 ms before the next shift. The sequence of fixated grating positions, which are numbered from 1 to 4, is indicated in the top row. (B) Top: Raster plot and firing rate profile, obtained as the peristimulus time histogram (PSTH), for a ganglion cell (Cell 1) under shifts from grating position 1 to position 2. The saccade-like transition period is marked by the shaded region. Bottom: Firing rate profiles of the same cell for all 16 transitions between the four grating positions. Data are displayed for 400 ms, following transition onset. For comparison, the considered time window is shown by a dashed rectangle in the PSTH above. The fixation positions before and after the saccade-like transition are denoted as 'starting position' and 'target position', respectively. (C) Response patterns of a sample cell (Cell 2) with sensitivity to image recurrence (IRS cell). Raster plots and PSTHs are shown for transitions to grating position 2, starting either from position 1 (top) or from position 2 (center). Bottom: Firing rate profiles of this cell for all 16 transitions. The IRS response is apparent by the post-transition firing rate peak for traces along the diagonal of this matrix (marked by small arrows). (D) Same as (C) for another sample cell (Cell 3) with sensitivity to image recurrence, but without a response during the transition itself.

detecting IRS cells; *Figure 2H*). For some of these cells, the first spike burst during the saccade was small or absent, as shown by the example of *Figure 1D*, yet after fixation onset, a spike burst still robustly marked those image transitions where starting and target position were equal.

All identified IRS cells showed strong similarity in their general response features. They were Off cells with transient responses to negative contrast steps and strongly biphasic spike-triggered averages under white-noise flicker of light intensity (*Figure 2B*). The biphasicness of the filters was particularly striking, and we quantified this by computing a biphasic index as the size ratio between the second filter peak and the first (*Zaghloul et al., 2007*; *Garvert and Gollisch, 2013*). This showed that, indeed, filters of identified IRS cells were systematically more biphasic than for other recorded cells (IRS cells: $1.04 \pm 0.40$, mean $\pm$ SD; other cells: $0.66 \pm 0.52$; $p<10^{-30}$; Wilcoxon rank-sum test). Furthermore, IRS cells showed fairly large receptive fields (*Figure 2C,E*; average diameter of $275 \pm 40$ μm, mean $\pm$ SD), and the spatial arrangement of receptive fields indicated that they border

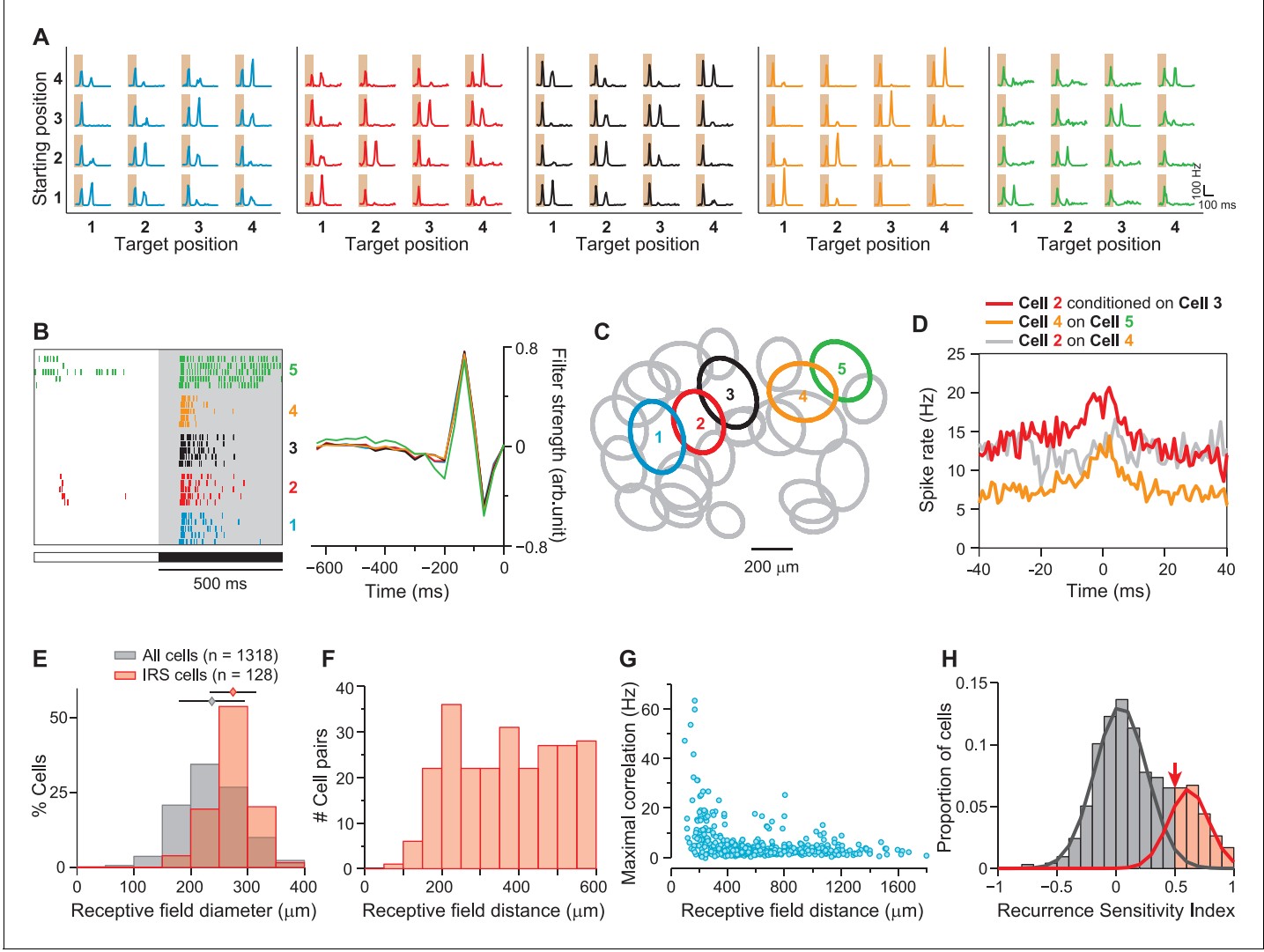

**Figure 2.** IRS cells show homogeneity in general response characteristics. (**A**) Responses of five simultaneously recorded sample IRS cells to the 16 saccade-like transitions. (**B**) Raster plots for repeated steps in light intensity (left) and temporal filters obtained as spike-triggered averages under full-field white-noise stimulation (right) for the five sample IRS cells. (**C**) Receptive field outlines of the five sample IRS cells (colored ellipses), together with all other receptive fields obtained in the same recording (gray ellipses). (**D**) Spike-timing correlations of spikes in pairs of IRS cells from among the five sample cells. (**E**) Distribution of receptive field sizes of IRS cells (red) from all recordings, compared to the distribution for all recorded cells (gray; 48 recordings). (**F**) Receptive field center distances of pairs of IRS cells. (**G**) Spike-time correlations of 456 pairs of IRS cells plotted against the receptive field center distance. (**H**) Recurrence Sensitivity Index (*RSI*) of 963 cells from 48 recordings included for analysis (see 'Materials and methods' for details). Cells with $RSI \geq 0.5$ were considered as IRS cells (red bars). The lines show a fit by a two-component Gaussian mixture model.

on each other with little overlap (*Figure 2C,F*), Moreover, neighboring cells displayed correlated spiking at short latencies of approximately 5 ms and smaller (*Figure 2D,G*), characteristic of gap-junction coupling (*Hu and Bloomfield, 2003*).

Together, these data suggest that the IRS cells identified in our recordings correspond to a single ganglion cell type. Moreover, the transient responses, biphasic filter shapes, and fairly large receptive field sizes of IRS cells match the functional characteristics of transient Off-alpha cells in mouse retina (*Huberman et al., 2008*; *van Wyk et al., 2009*; *Wang et al., 2011*; *Baden et al., 2016*). Indeed, we found, as discussed in more detail below, that IRS cells can be targeted in patch-clamp recordings by aiming for cells with the morphological characteristics of transient Off alpha cells (*Pang et al., 2003*; *Murphy and Rieke, 2006*; *Margolis and Detwiler, 2007*), indicating that it is this specific type of ganglion cells that constitutes the class of IRS cells.

## Influence of stimulus history on the encoding of image recurrence

The responses of IRS cells clearly display a strong dependence on stimulus history across the image transition; the same post-transition image can evoke fundamentally different response patterns, depending on what stimulus preceded the transition. To examine this dependence on stimulus history further, we applied several variations of the saccade stimulus. To test whether the actual motion during the saccade-like transition bears any significance for the characteristic responses of IRS cells, we randomly masked individual transition periods with uniform illumination of mean intensity. Responses of IRS cells to these 'blink-like' transitions were nearly indistinguishable from those to the saccade-like transitions (*Figure 3A*), indicating that the stimulus motion during the transition was not necessary for generating the IRS response and merely served to transiently blur the image.

We further tested how the response to an image recurrence compared to the response under presentation of the same grating in isolation, that is, following an extended period of mean gray illumination. As expected, IRS cells responded well to the flashed gratings at all four positions. Compared to the responses after fixation onset of a recurring grating, the responses to isolated gratings occurred with shorter latency, yet were generally weaker (*Figure 3B*). This is noteworthy because under the saccade-like stimulation, the prior exposure to the same grating should lead to adaptation and thereby decrease sensitivity to the recurrence of the grating. Furthermore, the spike burst that occurs during the transition itself and that precedes the spike burst at the onset of the new fixation by a mere 100–150 ms should trigger spike-frequency adaptation in the ganglion cell (*O'Brien et al., 2002*; *Kim and Rieke, 2003*) and thereby additionally decrease sensitivity when the grating recurs after the transition. Yet, despite these adaptation processes, the responses to recurring gratings were still stronger than for isolated gratings. This suggests that mechanisms beyond standard stimulus integration and response adaptation are involved in creating this non-standard history dependence.

Saccadic shifts in the direction of gaze may be accompanied by changes in low-level stimulus statistics, such as visual contrast. We therefore tested whether the sensitivity to image recurrence is affected by contrast changes. To do so, we randomly selected the contrast for each fixation to be either 30 or 60% and masked the transition periods by background illumination. We found that, regardless of the combination of contrast before and after the transition, post-transition spike bursts only occurred for equal starting and target position of the grating (*Figure 3C*), indicating a certain invariance to contrast in the detection of a recurring spatial stimulus layout by the IRS cells. The robustness of the characteristic IRS responses is also underscored by varying the duration of the transition. In fact, we found that the specific sensitivity to image recurrence for these cells occurred for transition durations from few tens to some hundreds of milliseconds (*Figure 3—figure supplement 1*). This indicates that the response to a recurring image does not result from an intrinsic oscillation mechanism that would resonate with the disappearance and reappearance of the image at some optimal interval.

## Dependence on spatial stimulus structure

The grating stimuli activate both receptive field center and surround with combinations of brightness and darkness. To dissect which aspects of the spatial stimulus structure are most relevant for controlling the sensitivity to image recurrence, we therefore manipulated the spatial structure of the saccade stimulus. First, we restricted stimulation to a small spatial patch, just slightly larger than the receptive field center, so that activation of the receptive field surround should be strongly reduced. This left the IRS responses essentially unaltered (*Figure 4A*), indicating that the relevant mechanisms operate inside the receptive field center.

Furthermore, the grating is composed of components with positive and negative contrast. To test whether the recurrence of one or the other of these components generates the IRS response, we decomposed the grating into its bright and dark stripes and presented transitions between patterns of only bright stripes as well as between patterns of only dark stripes. Here, we did not use actual saccadic shifts because these would simply correspond to a bright/dark grating with altered mean illumination and reduced contrast. Thus, we focused on transitions that were masked by background (gray) illumination. Unlike for the full grating transitions, which always combine brightening at some locations with darkening at others, some stimulus locations now remain unchanged at transition onset and/or fixation onset, and the other locations simultaneously all experienced either

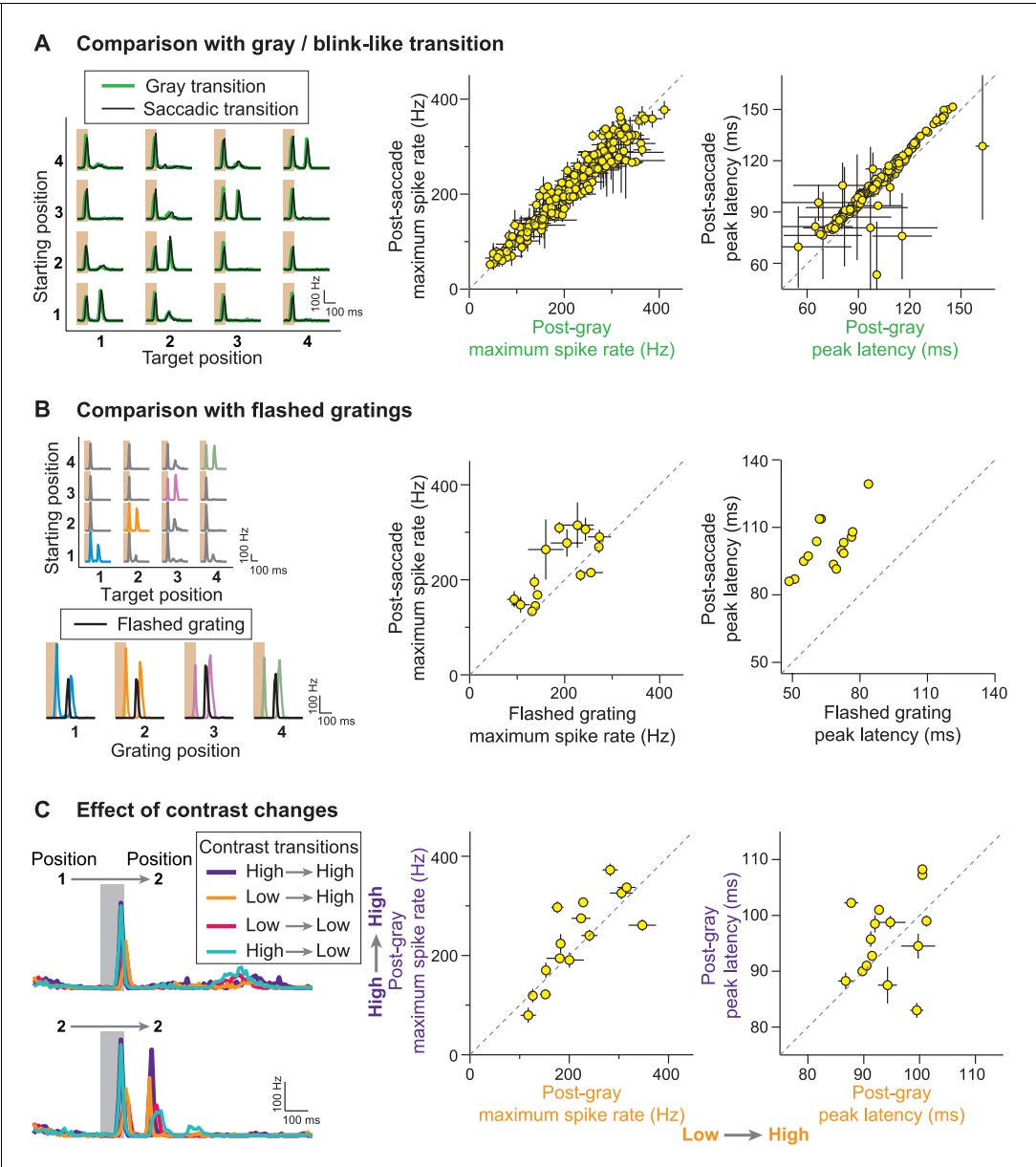

**Figure 3.** Stimulus history dependence of IRS responses. (**A**) Comparison of responses from a sample IRS cell (left) for saccade-like transitions (black traces) and transitions masked by a gray screen of mean illumination (green traces) as well as population analysis of the maximum firing rate and its latency for the two transition types (right). For the population analysis, each data point represents the average for a single cell over the four grating positions with equal starting and target position, and error bars denote standard errors, but are smaller than the symbol size for many data points (N = 209 cells from 48 recordings). (**B**) Comparison of responses from a sample IRS cell (left) for saccade-like transitions of a recurrent grating (colored traces) and for flashing the same grating in isolation (black traces) as well as population analysis as in (**A**) of the maximum firing rate and its latency (right; N = 15 cells from five recordings). The flashed gratings were preceded by mean-intensity illumination, and their responses were aligned to the onset of the new fixation in the saccade-like transitions. (**C**) Responses of a sample IRS cell (left) for transitions that included random changes in contrast between a high (60%) and low (30%) level as well as population analysis of the maximum firing rate and its latency as in (**A**) for the transitions to the high-contrast target images (right; N = 15 cells from five recordings).

The following figure supplement is available for figure 3:

**Figure supplement 1.** IRS responses are robust to variations of transition duration.

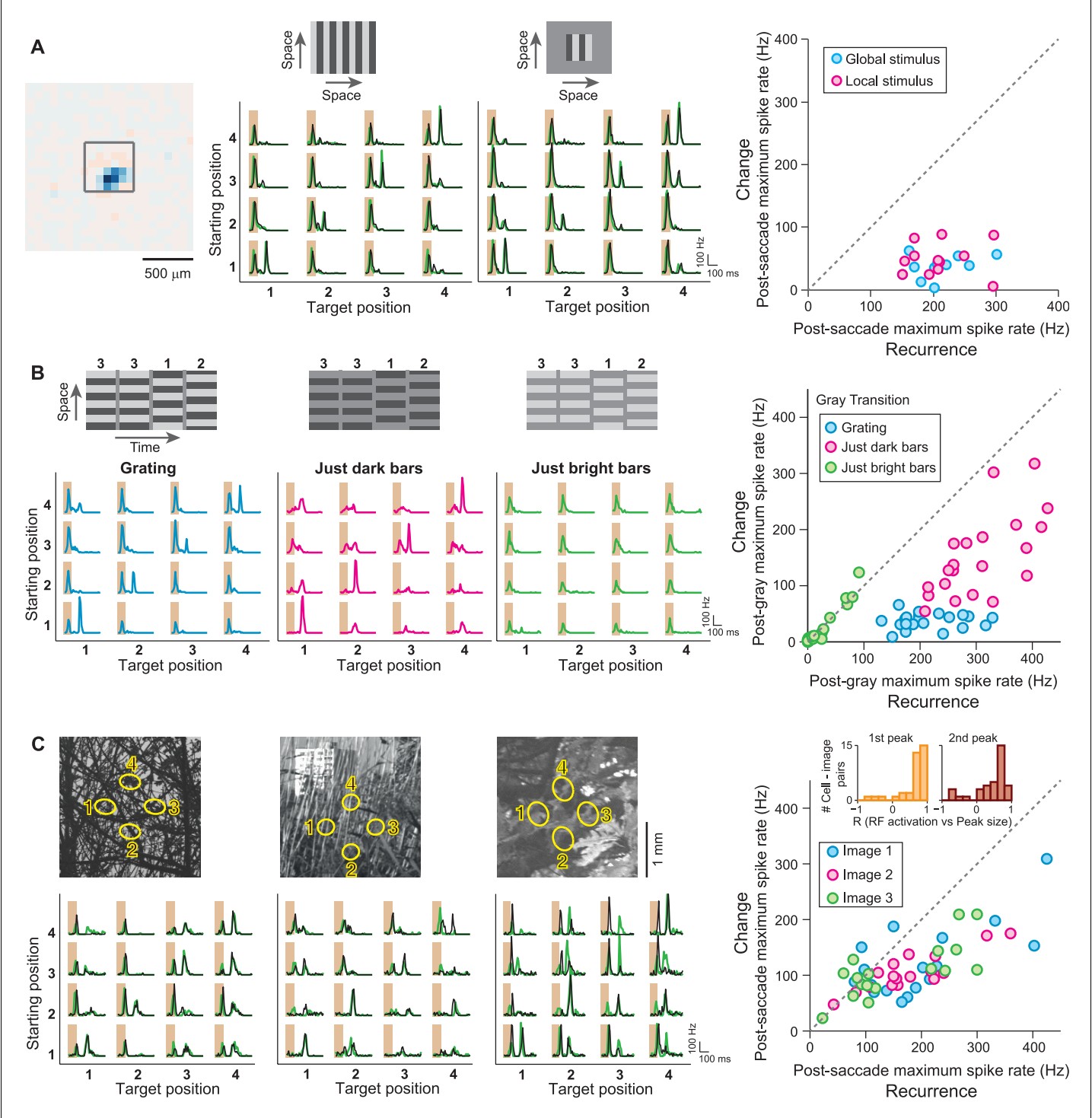

**Figure 4.** IRS responses are robust to spatial stimulus structure. (**A**) Influence of spatial extent of the stimulus. Left: Receptive field of a sample IRS cell. The outlined square shows the region of a local saccade stimulus. Middle: Firing rate profile of the sample IRS cell for the standard saccade stimulus, displayed over the entire screen, and for a smaller stimulation area of 480 µm x 480 µm. Responses under masked transitions by a gray screen are shown in green. Right: Population analysis (N = 12 cells from three recordings). For each recorded IRS cell, the maximum firing rate after onset of the target image was averaged over transitions with equal starting and target position ('Recurrence', x axis) and for the other transitions, where starting and target position differed ('Change', y axis). (**B**) Differential effects of bright and dark grating components. Left: Responses from a sample IRS cell to the standard grating stimulus and to gratings that contained only the dark or bright bars, all for transitions with a gray screen. The space-time representations of the stimuli are shown on top. Right: Population analysis (N = 21 cells from five recordings) as shown in (**A**). (**C**) Responses of three IRS

*Figure 4 continued on next page*

*Figure 4 continued*

cells (bottom, black traces) to saccade-like shifts of natural images (top). Ellipses in the images denote the receptive field outlines at the four fixation positions. Transitions lasted 100 ms and either occurred linearly between two fixation positions (starting position different from target position) or were composed of a shift to the center during the first 50 ms, immediately followed by a shift back to the starting position (equal starting and target position). For comparison, responses where the image was blurred during the transition are shown in green. Right: Population analysis of IRS cell responses to three natural images (N = 17 cells from four recordings). Inset: Distributions of correlation coefficients between the size of the first firing-rate peak and the integrated darkening in the receptive field at transition onset (left) as well as between the size of the second firing-rate peak and the integrated darkening in the receptive field at fixation onset (right).

The following figure supplement is available for figure 4:

**Figure supplement 1.** IRS responses are robust to variations of spatial stimulus scale.

brightening or darkening (see schematic stimuli in *Figure 4B*). We found that the dark bars alone were sufficient to trigger the characteristic IRS response, with strong post-transition peaks in the firing rate only when the target position matched the starting position of the dark bars (*Figure 4B*). Note, though, that transitions with a change in grating position now caused slightly increased post-transition responses as compared to stimulation with the full grating, indicating that the response suppression for a change in stimulus position was not as robust as for a true saccade-like transition where the mean stimulus intensity during transition and fixation matched. Transitions of only bright bars, on the other hand, did not evoke post-transition responses by themselves, but triggered the first response peak that occurred during the transition, indicating that this is a response to the offset of non-preferred stimulus contrast.

We also checked whether the characteristic IRS responses could also be evoked by saccade-like shifts of natural images. Here, natural photographs were shifted between different positions, and the recurrence of the same image position was achieved by translating the image rapidly forth and back, similar to the occurrence of a saccadic intrusion (*Abadi and Gowen, 2004*; *Otero-Millan et al., 2011b*). These image recurrences indeed generally led to the strongest post-transition responses (*Figure 4C*). Furthermore, similar to the grating stimuli, the responses of IRS cells to natural images were largely independent of whether the transition occurred in a saccade-like or blink-like fashion, with the motion masked by mean illumination. Note, though, that responses of individual cells for different fixation positions were somewhat more variable than for grating stimuli. This response variability is likely due to variability of stimulus patterns inside the receptive field; patches of fairly homogeneous bright illumination, for example, are not expected to yield strong IRS responses because of the Off-type nature of these cells.

To test this hypothesis, we computed, for each pair of recorded cell and applied image, the correlation between the receptive field activation and the size of the firing rate peak under image recurrence (inset of *Figure 4C*, see Materials and methods). For both transition onset and fixation onset, receptive field activation was here computed as the integrated amount of darkening, weighted by the profile of the receptive field. Indeed, we found systematically positive correlations both between the amount of darkening at transition onset and the size of the first response peak ($p=1.1 \cdot 10^{-6}$, Wilcoxon signed-rank test) as well as between the amount of darkening at fixation onset and the size of the second response peak ($p=2.3 \cdot 10^{-4}$). Thus, variations in receptive field activation by natural images appear to be a strong source of response variability. Note, however, that despite this response variability on the single-cell level, the population response of IRS cells can still robustly encode global image recurrences, with the strongest contributions to the population response generally coming from different subsets of IRS cells for different images.

Finally, the robustness of the IRS responses to changes in spatial stimulus structure is also reflected by the fact that the sensitivity to image recurrence occurred in a similar fashion for gratings with different spatial frequencies (*Figure 4—figure supplement 1*).

## Spatial sensitivity and potential role for fixational eye movements

We investigated the spatial sensitivity of the IRS cells by testing how different the pre- and post-transition images have to be in order to suppress the post-transition spike burst. Using a finer resolution of the applied grating positions, we found that net translations of about 30–50 μm of the grating

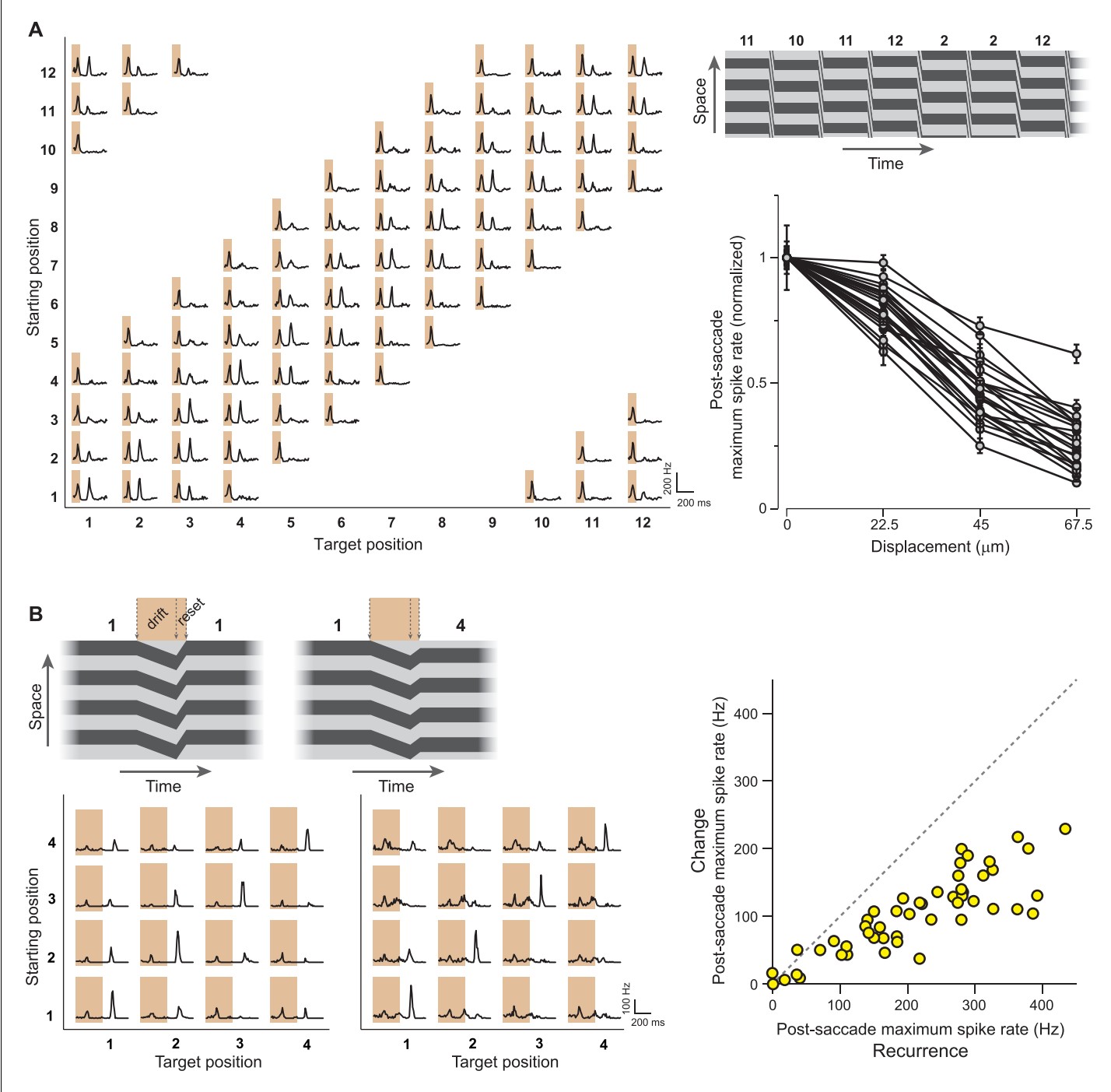

**Figure 5.** IRS responses are sensitive to small net translations and microsaccade-like image resets. (**A**) Left: Responses of a sample IRS cell to a stimulus with fine net displacements between starting and target position, corresponding to a resolution of 30° of grating phase or 22.5 μm translation on the retina. Top right: Space-time representation of the stimulus. Bottom right: Population analysis of the dependence of the averaged post-transition maximum firing rate on the net displacement between starting and target image (N = 22 cells from seven recordings). Firing rates were normalized to the value obtained for zero net displacement, and error bars correspond to standard errors. Lines connect data points of individual IRS cells. (**B**) Left: Responses of two IRS cells (bottom) to drift-reset stimulus motion, as schematically depicted on top, which can represent a correct reset (left) or a reset that is too short (right) or too far. Right: Population analysis of responses to the drift-reset stimulus (N = 50 cells from five recordings). DOI: 10.7554/eLife.22431.008

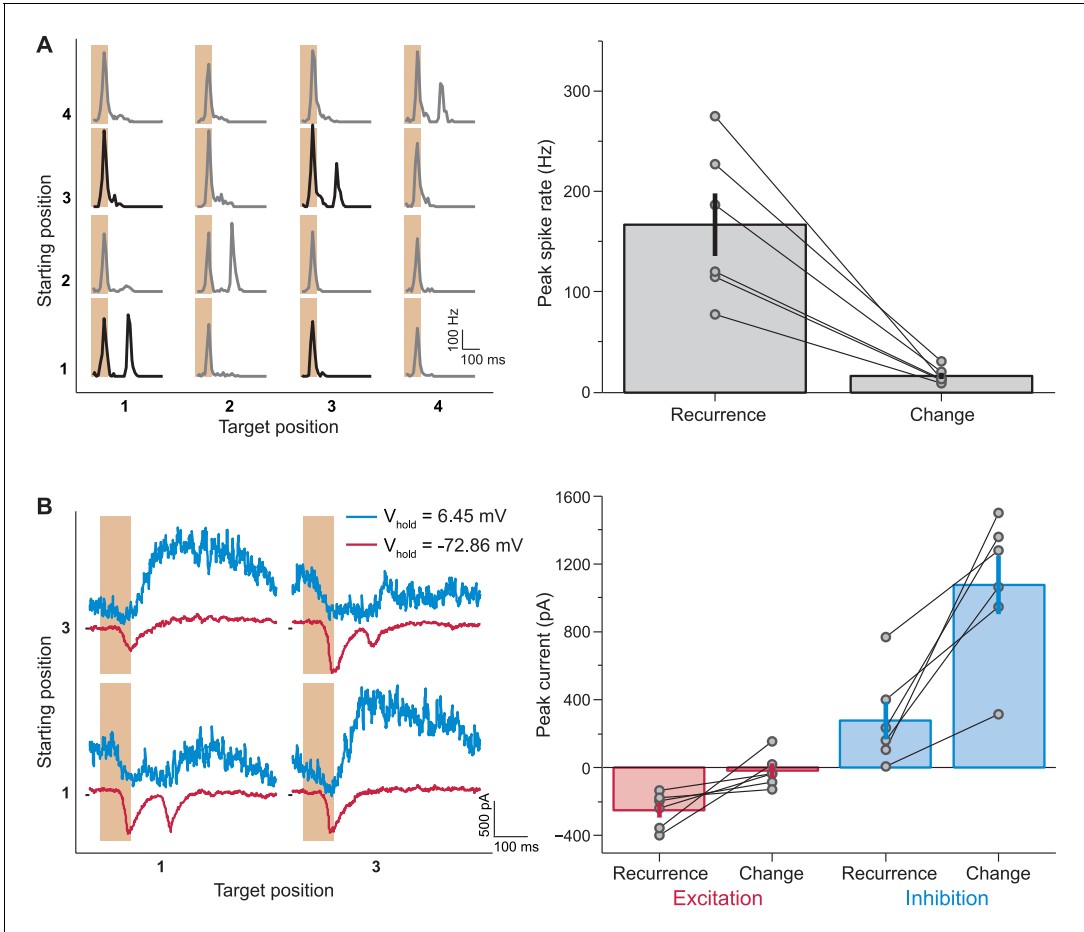

**Figure 6.** Synaptic inputs that mediate IRS responses. (**A**) Left: Responses of a sample IRS cell recorded in loose-patch mode. Right: Population analysis of the post-transition firing rate peaks of IRS cells recorded in loose-patch mode (N = 6 cells). (**B**) Left: Currents of the same sample cell as in (**A**), recorded in whole-cell mode. Note that here the stimulus is restricted to transitions between two fixation positions, corresponding to the data displayed in bold in (**A**). Note further that the sequence of fixation positions was not randomized in these experiments, but followed a fixed, alternating sequence of change and recurrence of grating position. This explains the slight differences in the level of inhibitory currents before the transition. Right: Population analysis of excitatory and inhibitory current responses. The excitatory peak current was determined as the minimal current in the window from 100 to 250 ms after fixation onset under a holding potential near −70 mV and the inhibitory peak current as the maximal current at the same time point under a holding potential near 0 mV. Error bars in (**A**) and (**B**) show standard errors.

were sufficient to reduce the IRS response by half (*Figure 5A*), indicating that the images are compared locally by the IRS cells on a scale that approximately matches typical bipolar cell receptive fields (*Berntson and Taylor, 2000*; *Schwartz et al., 2012*).

The observed sensitivity to fine spatial scales let us hypothesize that the IRS responses might play a role for stimulus translations that are smaller than full saccades. In fact, rapid recurrences of image position may be of particular relevance in the context of fixational eye movements because small drifts in eye position may be actively corrected by microsaccades to reset the proper direction of gaze (*Engbert and Kliegl, 2004*; *Rolfs, 2009*; *Otero-Millan et al., 2011a*; *Costela et al., 2014*). We therefore tested the responses of IRS cells to stimuli that were composed of a small drift-like translation for half a grating period, followed by a rapid reset that either correctly compensated for the drift or was too far or too short. Here, IRS cells responded with a distinct spike burst after the reset, which was most pronounced when the target position after the reset was equal to the starting position before the drift (*Figure 5B*). Thus, the responses of IRS cells to combinations of drift and reset marked correct reset sizes that compensated for the drift.

## Inhibitory interactions underlie the sensitivity to image recurrence

In order to elucidate the mechanism behind the unexpected response patterns of IRS cells, we aimed at recording intracellularly from these cells. Based on the general response characteristics of IRS cells (*Figure 2*), we had hypothesized that these cells corresponded to transient Off-alpha cells, which can be targeted for patch-clamp recordings via their large soma size. Indeed, cell-attached recordings of spiking activity from such candidate cells showed the same sensitivity to image recurrence as we had observed in the multielectrode-array recordings (*Figure 6A*). This allowed us to establish whole-cell recordings from identified IRS cells and assess the excitatory and inhibitory inputs during image transitions (*Figure 6B*).

When the grating after the transition appeared with altered spatial phase, the cells received strong inhibitory input. This inhibition was suppressed when the grating reappeared with the same spatial phase (peak currents 280 pA ±110 pA and 1070 pA ±170 pA, mean ± SEM, for recurrence and change, respectively; p=0.0054, paired *t*-test). A corresponding dependence on image recurrence was observed for the excitatory input. Here, a second response peak occurred only when the grating reappeared with the same spatial phase. When the grating phase changed across the transition, this excitatory input was suppressed (peak currents −249 pA ± 43 pA and −18 pA ± 41 pA, mean ± SEM, for recurrence and change, respectively; p=0.031, paired *t*-test; *Figure 6B*). Thus, both inhibition and excitation depend on the stimulus transition in a way that supports strong responses for recurring images. This parallel modulation of excitation and inhibition may be explained by a single source of inhibition, which acts both directly onto the ganglion cell as well as onto the bipolar cell terminals that mediate the excitatory synaptic input to the ganglion cell. This inhibition would be active after a transition with a change in grating position and thereby provide inhibitory input to the ganglion cell and simultaneously suppress the excitatory input. Such dual effects of inhibitory signals are well known; AII amacrine cells, for example, project both directly onto transient Off-alpha cells as well as onto their presynaptic bipolar cell terminals (*Bloomfield and Dacheux, 2001*; *Ivanova et al., 2006*; *Manookin et al., 2008*; *Murphy and Rieke, 2008*).

To further probe the role of inhibition in generating the IRS responses, we pharmacologically blocked either glycinergic or GABAergic inhibition and evaluated the effect on the responses under saccade-like stimulus transitions. When we applied the glycine receptor antagonist strychnine, we found that IRS cells lost their specificity in responding to recurring images. Instead, the cells responded non-specifically to all fixations under strychnine (*Figure 7A*). By contrast, when we applied the GABA$_A$-antagonist gabazine to the retina, we found that the sensitivity to image recurrence was also abolished, but now because the response for the recurring image was suppressed (*Figure 7B*). Thus, blocking glycinergic inhibition and blocking GABAergic inhibition had opposite effects, with the former leading to an increase of activity after change across the transition and the latter to a suppression of activity after image recurrence. Together, this suggests that glycine-mediated inhibition acts directly onto the ganglion cell or onto its excitatory input, whereas GABAergic inhibition has an indirect effect because the activity reduction upon its block indicates a sign-inverted effect of this inhibitory signal. This lets us hypothesize that the presynaptic circuit of IRS cells involves serial inhibition, with glycinergic inhibition acting directly on the ganglion cell activity and GABAergic inhibition acting on the source of the glycinergic inhibition.

Indeed, transient Off-alpha cells receive strong glycinergic inhibition (*Zaghloul et al., 2003*; *Murphy and Rieke, 2006*; *Margolis and Detwiler, 2007*; *van Wyk et al., 2009*), for example through AII amacrine cells (*Manookin et al., 2008*; *Murphy and Rieke, 2008*; *Münch et al., 2009*), which themselves appear to receive GABAergic inhibition (*Marc et al., 2014*; *Zhang et al., 2014*). Thus, the hypothesized circuit of serial inhibition exists at least in principle. Yet, pharmacological interventions in complex neuronal networks, such as the retina, must be interpreted with care because of the possibility of off-target effects. Thus, to investigate whether, indeed, the hypothesized serial inhibition motif provides a viable mechanism for generating the characteristic responses of IRS cells, we explored whether they could be reproduced by a simple model circuit.

## A circuit model with local serial inhibition explains sensitivity to image recurrence

We set up a simple computational model of an IRS cell (*Figure 8A,B*), which processes the visual stimulus in localized spatial subunits, each containing Off-pathway excitation and (glycinergic) On-

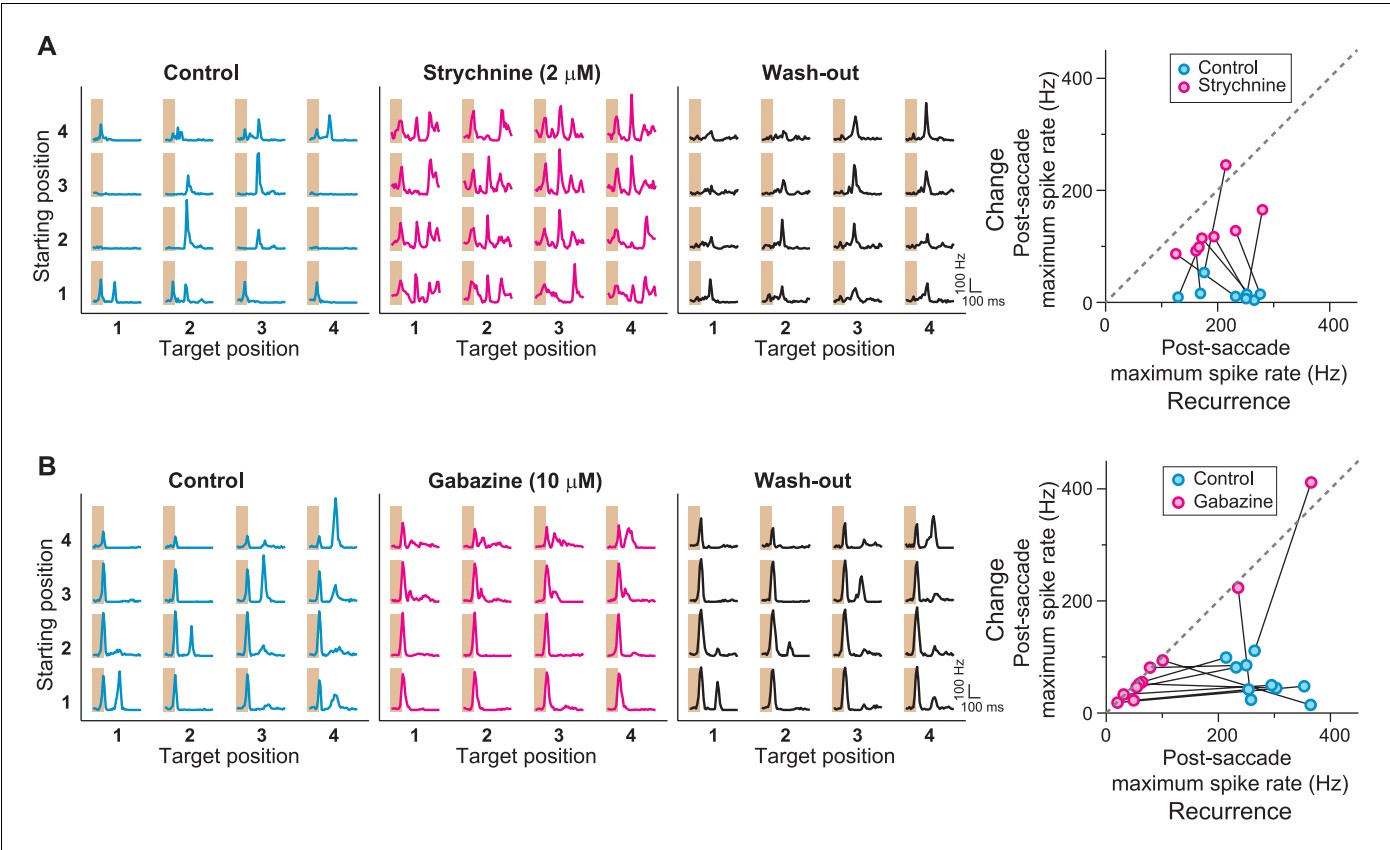

**Figure 7.** Role of GABA_A- and glycine-mediated inhibition in generating IRS responses. (**A**) Left: Firing rate profile of a sample IRS cell before, during, and after application of the glycine blocker strychnine. Right: Comparison of responses before (blue) and during drug application (magenta) for all recorded IRS cells. For each cell and condition, the maximum firing rate after onset of the target image was averaged over transitions with equal starting and target position ('Recurrence', x axis) and for transitions with different starting and target positions ('Change', y axis). Connected data points come from the same cell (N = 8 cells from three recordings). (**B**) Firing rate profiles of a sample IRS cell before, during, and after application of the GABA_A blocker gabazine. Right: Comparison of responses before (blue) and during (magenta) drug application for all recorded IRS cells as in (**A**) (N = 10 cells from three recordings). The two cells with strong spike rate increases under gabazine for transitions of the 'Change' type displayed a strong broadening of the first response peak into the post-transition analysis window, but did not show distinct second response peaks.

pathway inhibition (see 'Materials and methods' for the details of the model implementation). The On-pathway inhibition is slow so that it can mediate an effect across the transition, but it can itself be transiently suppressed by (GABAergic) inhibition from the Off pathway.

We found that this model is able to faithfully reproduce the sensitivity to image recurrence across saccade-like grating shifts (*Figure 8D*). To gain insight into how the IRS responses arise from this model, *Figure 8C* displays the activation of some of the local interneurons for different local stimuli with either a net change of luminance across the transition (left) or a recurrence (right). For stimuli with local changes across the transition, this shows that locations with a net darkening lead to two peaks in the excitation mediated by the Off-type bipolar cell, coming from the offset of the previous fixation and the onset of the new fixation, respectively. Locations with net brightening, however, provide slow, strong, long-lasting inhibition through the activity of the On-type amacrine cell, which is triggered by both transition onset and fixation onset. This inhibition suppresses activity after fixation onset, but, owing to the low-pass filtering, is too slow to suppress the earlier activity peak during the transition.

For image recurrences, however, no location provides strong, long-lasting inhibition, because the On-type amacrine cell is either suppressed by transition onset ('Bright-to-bright' locations) or by fixation onset ('Dark-to-dark' locations). Thus, here, the first excitation peak, generated by transition onset at 'Bright-to-bright' locations, leads to spiking because it escapes the slower inhibition, and

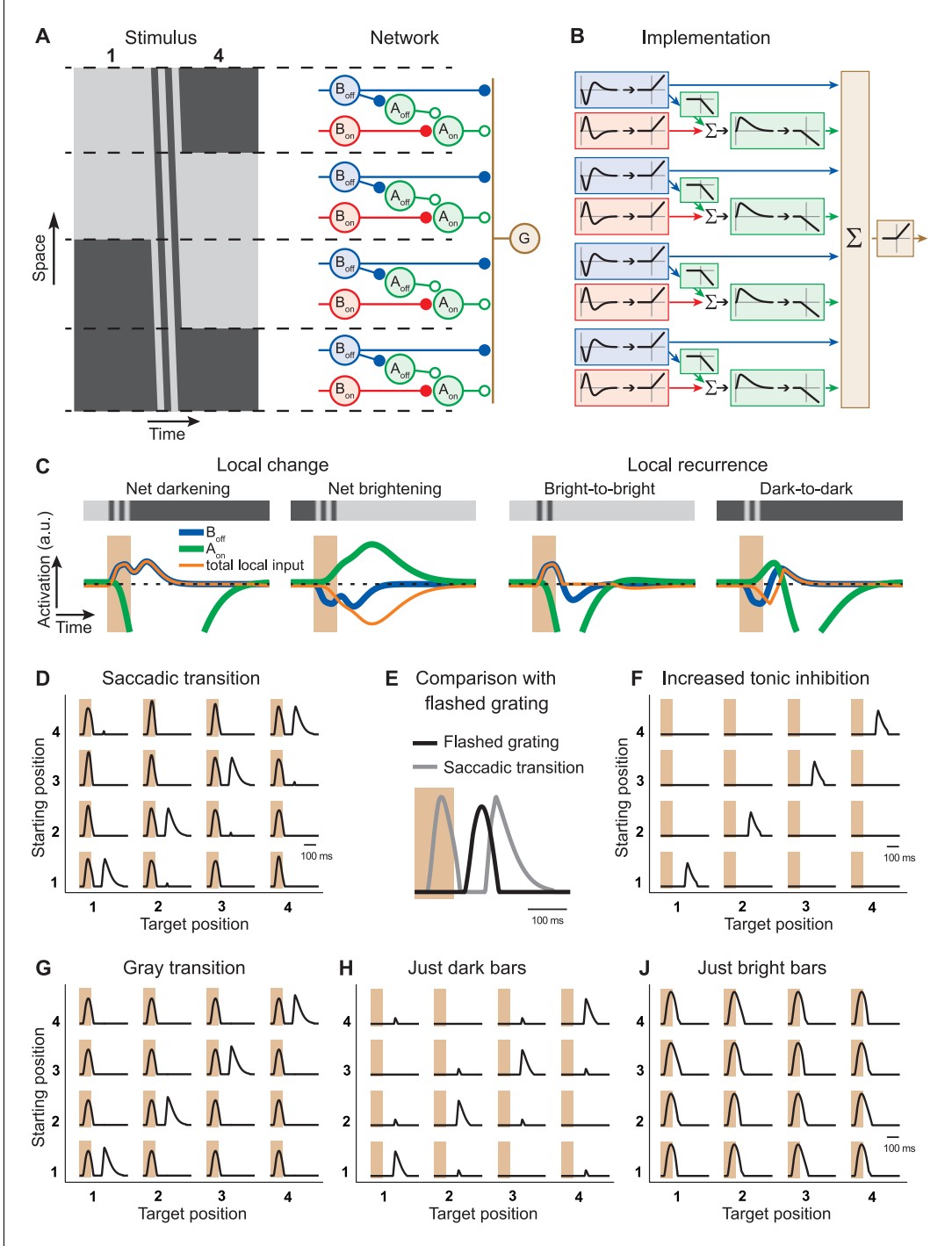

**Figure 8.** Computational model with local serial inhibition explains IRS responses. (**A**) Schematic depiction of stimulus and model circuit. Left: Space-time representation of the stimulus for a sample transition from grating position 1 to 4. Right: Model circuit. The ganglion cell (G) receives excitatory input from Off-type bipolar cells ($B_{off}$) and inhibitory input from On-type amacrine cells ($A_{on}$), which are activated by On-type bipolar cells ($B_{on}$). In addition, these amacrine cells receive inhibition from Off-type amacrine cells ($A_{off}$), which are activated by the Off-type bipolar cells. (**B**) Functional depiction of the model circuit. Bipolar cells are represented by temporal filters, shown here in the time domain, followed by a half-wave rectifying nonlinearity. Amacrine cells implement similar rectification, together with sign-inversion, and the $A_{on}$ cell additionally applies a temporal low-pass filter and a non-zero threshold of rectification, accounting for tonic inhibition received by the ganglion cell. The ganglion cell is represented by a sum ($\Sigma$) of its inputs and another rectification stage. (**C**) Model responses of local interneurons. Activation curves of Off-type bipolar cells (blue) and On-type amacrine cells (green) are displayed for a saccade-like transition of the grating. The 100-ms transition is marked by the shaded areas. The corresponding local light intensity is indicated in the top row. Also shown is the resulting total input to the ganglion cell for each local stimulus
*Figure 8 continued on next page*

*Figure 8 continued*

(orange), which is obtained as the rectified $B_{off}$ activity minus the rectified $A_{on}$ activity. The activation curves of the On-type bipolar cells and Off-type amacrine cells are not displayed because they differ from those of the Off-type bipolar cells only by a sign inversion and a threshold, respectively. (D) Simulated firing rate profiles for the 16 saccade-like transitions. (E) Comparison of the model responses to a recurring grating and to the same grating flashed in isolation, analogous to the experimental data shown in *Figure 3B*. (F) Simulated firing rate profiles for the 16 saccade-like transitions with increased tonic inhibition supplied by the amacrine cell $A_{on}$, reproducing data where the spike burst during the transition itself was absent (cf. *Figure 1D*). (G) Simulated firing rate profiles for gray transitions (cf. *Figure 3A*). (H, J) Simulated firing rate profiles for stimuli with either only dark or bright bars (cf. *Figure 4B*).

The following figure supplements are available for figure 8:

**Figure supplement 1.** Responses from simulations of the IRS cell model under different transition durations.

**Figure supplement 2.** Responses from simulations of the IRS cell model under variations of parameters.

the second excitation peak, generated by fixation onset at 'Dark-to-dark' locations, leads to spiking once it overcomes the decaying On-type inhibition.

These dynamics can also explain why the responses after an image recurrence are somewhat stronger but occur with a longer latency than the responses to isolated image flashes (*Figure 8E*), as experimentally observed (cf. *Figure 3B*). The isolated flash of a grating triggers both excitatory and inhibitory inputs to the ganglion cell. Because excitation is faster than inhibition (in the model reflected by the low-pass filtering of the On-type amacrine cell), a spike burst is generated at short latency, but the peak response is then already suppressed by the rising inhibition. For an image recurrence, on the other hand, inhibitory inputs are present during the early phase of the new fixation (see *Figure 8C*, 'Dark-to-dark'), preventing short-latency spiking. But eventually, this inhibition is strongly suppressed for both 'Bright-to-bright' and 'Dark-to-dark' locations, allowing a relatively strong peak in the firing rate, but with longer latency.

The model also incorporates the experimentally observed variability in the occurrence of a first response peak during the saccade (cf. *Figure 1C,D*) by just varying a single parameter, namely the amount of tonic inhibition onto the ganglion cell (*Figure 8F*). Increased tonic inhibition can prevent the first response peak but not the IRS response after fixation onset because it becomes itself suppressed through the serial inhibition. Responses to other variants of the stimulus transitions are furthermore reproduced without any adjustments of parameters. Model responses to transitions with masked saccades ('blink-like' transitions, *Figure 8G*) were nearly identical to those under saccade-like stimulation (cf. *Figure 8D*). And model simulations under stimulation with dark bars only (*Figure 8H*) showed IRS responses similar to experimental results (cf. *Figure 4B*), including the increased small responses under changing stimulus position, whereas responses to stimulation with just bright bars generated the first response peak only (*Figure 8J*).

Note that the presented model was intended for checking whether a serial-inhibition circuit can qualitatively reproduce the characteristics of the IRS cell responses, not as a quantitative description of firing rate profiles, and we therefore did not perform any systematic parameter optimization. Nonetheless, it is instructive to check how the model performs under different variations of stimulus and model parameters. When applying the model to transitions of different durations (*Figure 8—figure supplement 1*), we observed that IRS-like responses occurred for transition durations as short as 33 ms and that longer transitions led to a gradual decline of the IRS response patterns and the emergence intermediate activity peaks, qualitatively similar to the experimental data (cf. *Figure 3—figure supplement 1*). There are, however, substantial quantitative deviations of the model responses from the experimental data for short and long transitions, in particular for saccade-like transitions, though less so for masked transitions. These indicate shortcomings of the model in its ability to resolve fast stimulus components, such as the very brief transitions or the stimulus fluctuations during the long transition period for long transitions. Yet, these deviations are not surprising, given that temporal stimulus integration in the model was implemented purely through filtering operations and neglects, for simplicity, well-known feedback and gain control operations (*Berry and Meister, 1998*; *Keat et al., 2001*; *Pillow et al., 2005*; *Jarsky et al., 2011*; *Ozuysal and Baccus, 2012*; *Liu and Gollisch, 2015*).

Finally, we investigated the robustness of the model to parameter variations (*Figure 8—figure supplement 2*). In particular, this shows that the model responses are largely invariant to whether the On-type inhibition goes directly to the ganglion cell or whether it acts presynaptically on the terminals of the Off-type bipolar cells, that is, before rectification of the excitatory input to the ganglion cell takes place. Furthermore, this analysis indicates that the qualitative behavior of the model does not depend sensitively on individual parameters, such as the strength of the On-type or Off-type inhibition or the time scales of the filters.

## Discussion

Here we showed that a specific subset of ganglion cells in the mouse retina displays an unexpected, non-standard dependence on stimulus history when stimulated with rapid transitions of images, as occur during dynamics of the direction of gaze, such as saccades or fixational eye movements. Rather than primarily responding to an increase in preferred contrast across a transition, these cells fire a distinct spike burst when the spatial pattern inside the receptive field is nearly identical before and after the transition (*Figures 1* and *2*). This encoding of image recurrence is robust to changes in transition type, visual contrast, and spatial structure of the image (*Figures 3* and *4*) and is exquisitely sensitive to changes in the image on a scale of few tens of micrometers, suggesting a potential role in monitoring fixational eye movements (*Figure 5*). Finally, a combination of intracellular recordings (*Figure 6*), pharmacology (*Figure 7*), and computational modeling (*Figure 8*) indicated that these responses are controlled by a circuit of serial inhibition that is triggered by the rapid image transition.

### Cell type identity of IRS cells

Cells that displayed the characteristic IRS response in the multielectrode-array recordings all showed similar general response characteristics; they were transient Off cells with a fairly large receptive field. Patch-clamp recordings from transient Off-alpha cells, which can be identified by their soma size and characteristic visual responses to flashed and flickering spots and gratings (*Pang et al., 2003*; *Murphy and Rieke, 2006*; *Margolis and Detwiler, 2007*; *Manookin et al., 2010*), corroborated our hypothesis that this cell type largely constitutes the class of IRS cells.

Transient Off-alpha cells in mice are known to project to both the dorsal lateral geniculate nucleus and the superior colliculus (*Huberman et al., 2008*; *Hong et al., 2011*), which may indicate that their responses contribute to multiple visual functions. Interestingly, the superior colliculus is known to be involved in the control of saccades, with at least a rough match in the topographic organization between the visual inputs and the saccadic directions (*Wang et al., 2015*). Thus, inputs from transient Off-alpha cells to the superior colliculus might provide sensory feedback to the motor control of gaze shifts and thereby contribute to establishing and monitoring the appropriate topographic organizations, which indeed appear to depend on visual experience (*Wang et al., 2015*).

We cannot exclude that other types of ganglion cells, perhaps more rarely recorded with the multielectrode arrays, also show similar response patterns and therefore contribute to our analysis of the class of IRS cells. Given the overall homogeneity of the response characteristics of IRS cells, however, we do not expect a large contribution from other cell types. In fact, the particular response to image recurrence was so distinct that IRS cells could often be readily detected during the experiment by a glance at the voltage trace on the oscilloscope. Thus, stimuli with rapid image transitions may represent an interesting complement to the standard stimulus repertoire for defining signature response characteristics and distinguishing cell types in physiological experiments (*Carcieri et al., 2003*; *Farrow and Masland, 2011*; *Baden et al., 2016*).

Yet, some variability between IRS cells was observed in the occurrence of a first response peak during the transition itself (*Figure 1C,D*). We suspect that this results from variability within a single cell type rather than from contributions of different types, in particular because the simulations showed that the first response peak sensitively depends on the level of tonic inhibition onto the ganglion cell (*Figure 8F*). Indeed, the level of tonic inhibition received by alpha ganglion cells can vary considerably, for example depending on the light level (*van Wyk et al., 2009*; *Marco et al., 2013*), and it thus appears likely that variability in tonic inhibition occurs between experiments and even between cells in individual experiments.

## Consequences for signal processing in the retinal network

Our results show that the rapid image transitions induced by natural dynamics of the eye can trigger interactions in the retinal network that provide retinal ganglion cells with previously unknown response characteristics. The serial-inhibition circuit that underlies the responses of IRS cells illustrates the computational power of the interconnected inhibitory network in the retina. Here, it operates by gating the inhibitory signal flow through the retina. Other examples of transient, context-dependent gating of retinal signal flow by serial inhibition have been found in the salamander retina (*Roska et al., 1998*; *Geffen et al., 2007*), suggesting that signal gating may be a general function of the inhibitory interconnections in the retina. Moreover, such gating of inhibitory signaling might be a general motif of adjusting signal processing to stimulus context; it was recently shown, for example, that the strength of cortical surround suppression depends on contextual image statistics (*Coen-Cagli et al., 2015*).

The deduced circuit model of the IRS cells postulates that the inhibitory interactions be sensitive to local stimulus structure on a scale much smaller than the receptive field. Sensitivity to small stimuli of few tens of micrometers has also been observed in the disinhibitory signals of sustained Off-alpha cells (*Marco et al., 2013*). The local nature of the inhibitory interactions suggests that the serial inhibition is either mediated by narrow-field amacrine cells or involves local processing in dendritic compartments of a larger amacrine cell (*Grimes et al., 2010*).

Investigating the IRS cells in the context of rapid image transitions was essential for discovering the interactions in the inhibitory circuit. While the offset of the pre-transition image triggers the slow and sustained direct inhibition onto the ganglion cell, the onset of the new image evokes the rapid transient suppression of this direct inhibition. Commonly used simple stimuli, such as isolated images or white-noise flicker, will therefore not effectively activate this interaction. Thus, rapid image transitions trigger a mode of operation in the retina that differs from the processing of simpler stimuli. It will be interesting to see whether similar operational switches can be found in other circuits or under other naturalistic stimulus contexts, thereby potentially allowing individual ganglion cells to fulfill different visual tasks in a stimulus-context-dependent fashion. Furthermore, one may expect that comparable studies of other cell types under stimulation with rapid image transitions or with similar naturalistic stimulus dynamics will help elucidate further features of the inhibitory interactions in the retinal circuitry.

## Consequences for neural coding

The characteristic IRS responses provide a distinct and sensitive code for the reappearance of spatial stimulus patterns. This explicit representation of image recurrence may be useful for different purposes of visual processing. For example, it may help distinguish and monitor different types of eye dynamics. While saccades are expected to alter visual input at many locations on the retina, saccadic intrusions, which typically occur as a coupled pair of saccades in opposing directions (*Abadi and Gowen, 2004*; *Otero-Millan et al., 2011b*), and resetting microsaccades may bring the same image back in focus within tens or hundreds of milliseconds. Microsaccades, for example, are hypothesized to correct for fixational-eye-movement drift (*Engbert and Kliegl, 2004*; *Rolfs, 2009*; *Otero-Millan et al., 2011a*) and for blinking-induced gaze shifts (*Costela et al., 2014*), and the monitoring of proper resets may aid the formation and maintenance of these corrective eye movements. Note, though, that such complex gaze shift patterns have so far primarily been analyzed in humans and other primates, and although the existence of saccadic eye movements in mice is now well established (*Sakatani and Isa, 2007*; *Keller et al., 2012*; *Wang et al., 2015*), detailed observations of gaze movement patterns in freely behaving mice, which may move their eyes as well as their heads, are still lacking.

In the context of saccades, the IRS responses may furthermore allow tagging those parts of the visual field where the stimulus pattern stays relatively constant. This may occur, for example, when shifting the gaze along a contour, which appears to be a preferred way of directing saccades (*Foulsham et al., 2008*). In this context, it is worth noting that the IRS spike burst typically occurs with a latency of more than a hundred milliseconds after the end of the transition, and it is thus well placed to escape any saccadic suppression occurring downstream from the retina (*Ibbotson and Krekelberg, 2011*).

Interestingly, we did not find evidence for an On-type ganglion cell with IRS responses. Although, again, we cannot exclude that On-type IRS cells do exist as a counterpart to the Off cells characterized here and were missed because of recording bias, one may wonder whether there is an advantage of detecting image recurrence based on negative rather than positive contrast in the image. One reason may be that negative contrast is encountered more frequently than positive contrast in natural scenes (*Ratliff et al., 2010*), which suggests a higher level of spatial detail available through the Off pathway.

From a more general perspective, the sensitivity to image recurrence can be viewed as a complement to the more common sensitivity to changes in the visual scene. This is analogous to other dual pathways of the visual system, such as the segregation into On and Off channels or the occurrence of adapting as well as sensitizing ganglion cells after a change in visual contrast (*Kastner and Baccus, 2011*). The occurrence of such dual information channels has been linked to the need for efficiently covering the natural range of a given stimulus feature (*Gjorgjieva et al., 2014*). Another counterpart to the sensitivity to change is formed by suppressed-by-contrast ganglion cells, which display response suppression for most changes in visual input and which have recently been implicated in the detection of image transitions (*Tien et al., 2015*). Yet, while the suppressed-by-contrast cells display suppression fairly unselectively to nearly any stimulus transition, the IRS responses depend in a specific fashion on the net change in the visual pattern across a transition. Also, the IRS responses to the drift-reset stimulus (*Figure 5B*) bear some resemblance to the synchronized activity of ganglion cells under reversal of motion direction (*Schwartz et al., 2007*). However, the latter appears to provide a general signal for the occurrence of a motion reversal and does not rely on inhibitory pathways (*Chen et al., 2014*), whereas the IRS response depends on the precise relation of image positions across a brief transition.

Beyond the retina, several studies have shown that naturalistic stimulus statistics in general and eye-movement-like stimulus dynamics in particular can influence the response statistics and tuning properties of neurons in visual cortex (*David et al., 2004*; *Kayser et al., 2004*; *Ruiz and Paradiso, 2012*; *Baudot et al., 2013*; *Coen-Cagli et al., 2015*). The present study shows that such fundamental influences of stimulus dynamics already occur in the retinal circuit and may thereby contribute to differences in processing of simple and natural stimuli in higher visual areas.

## Materials and methods

### Tissue preparation

All experimental procedures were performed in accordance with institutional guidelines of the University Medical Center Göttingen, Germany. Retinas were isolated from wild-type mice (C57BL/6) of either sex. Before recordings, mice were dark-adapted for at least 30 min and sacrificed by cervical dislocation. Eyes were enucleated quickly, hemisected, and kept in oxygenated (95% $O_2$ and 5% $CO_2$) Ames' medium, buffered with 22 mM $NaHCO_3$ to maintain pH 7.4. Isolation of the retina from the eyecup was performed in a darkroom with infrared illumination under a stereo-microscope that was equipped with a pair of night vision goggles. In some cases, dim red light (wavelength above 690 nm) and no night vision goggles were used for preparation.

### Multielectrode-array recordings

Spike trains of retinal ganglion cells were recorded extracellularly with planar multielectrode arrays (Multichannel Systems). The isolated retinas were cut in half and placed ganglion cell-side-down on an array of either 252 or 60 electrodes (10 or 30 µm electrode diameter; 60 or 100 µm minimal electrode distance). During recordings, retinas were continuously perfused with the oxygenated Ames' medium at approximately 5 ml/min. The bath solution was heated to a constant temperature around 33–34°C via an inline heater in the perfusion line and a heating element below the array. Bath temperature was monitored by a thermal sensor inserted into the bath chamber. In some experiments, gabazine (SR95531; 10 µM; Sigma-Aldrich) or strychnine (2 µM; Sigma-Aldrich) was temporarily added to the perfusion.

Spike times of individual ganglion cells were determined from the recorded voltage traces by a custom-made spike sorting program, based on fitting the voltage-trace segments of potential spikes with a Gaussian mixture model (*Pouzat et al., 2002*). Only units with well separated clusters of

voltage-trace segments and with a clear refractory period were used in the analysis. Visual stimuli were generated by custom-made software, written in C++ and OpenGL, presented in the photopic range on a gamma-corrected cathode-ray-tube monitor (either 5.1 or 9.1 mW/m$^2$ mean intensity, 100 Hz refresh rate, 6 µm pixel size on the retina) or on a gamma-corrected miniature white-light oLED monitor (eMagin; 800 × 600 pixels; 3.3 mW/m$^2$ mean intensity; 60 Hz refresh rate, 7.5 µm pixel size on the retina), and focused on the photoreceptor layer of the retina with standard optics.

## Patch-clamp recordings

For loose-patch and whole-cell recordings, isolated retinas were attached to nitrocellulose filter membrane so that they covered a hole in the filter of 1 mm diameter. The retina was transferred into a glass-bottom recording chamber on an upright microscope and perfused with 5 ml/min oxygenated Ames' medium, heated to 33°C. Retinal ganglion cells were targeted in the area over the hole in the filter membrane with borosilicate glass microelectrode pipettes (resistance 3–5 MOhm) under infrared illumination, monitored through a CCD-camera. The pipette solution for loose-patch recordings was oxygenated Ames' medium, and a caesium-based intracellular solution was used for whole-cell recordings (120 mM Cs-MeS0$_3$, 3 mM NaCl, 3 mM QX314, 5 mM BAPTA, 5 mM EGTA, 10 mM HEPES, 2 mM MgATP, 0.3 mM NaGTP, 1 mM Lucifer Yellow CH, titrated to pH 7.5 at room temperature with TEA-OH). Visual stimuli were generated with the same software as for the multielectrode-array recordings and projected onto the retina through the condenser lens with a gamma-corrected DLP-pocket-projector (M109S, Dell; 16.5 mW/m$^2$ mean intensity; 60 Hz refresh rate; 800 × 600 pixels; 2.5 µm pixel size on the retina), equipped with a near-UV LED (emission peak at 390 nm).

We targeted retinal ganglion cells with large, round somas (*Pang et al., 2003*; *Murphy and Rieke, 2006*; *Margolis and Detwiler, 2007*). Transient Off-alpha cells were identified by their spiking responses to contrast steps in circular regions, centered on the cell soma, and to reversals of fine spatial gratings (*Manookin et al., 2010*). Specifically, we only included cells for further analysis that showed transient responses to negative-contrast spots as well as clear responses to fine reversing gratings and to high-frequency contrast reversals of spots up to 9 Hz. For estimating synaptic conductances in whole-cell recordings, we performed voltage-clamp recordings at eight different nominal holding potentials (−75 to +35 mV). Capacity and series resistance compensation were set to 80%. Typical series resistance was 9–25 MOhm, and the targeted ganglion cells generally had a membrane resistance of 30–50 MOhm. Individual recordings with high series resistance and low membrane resistance were discarded when the series-resistance-induced error on the holding potential frequently exceeded 10% (*Manookin et al., 2010*). All recordings were corrected for pipette offset and for the liquid-junction-potential of −15.6 mV. For each stimulus, recorded current traces were averaged over 3–5 repeats at each holding potential, and leak currents were subtracted, based on the measured current under homogeneous illumination at mean intensity, obtained between stimulus repeats. From this, we obtained excitatory current traces recorded with holding potential close to −70 mV (average ± SD: −76.0 mV ± 3.9 mV), and inhibitory current traces with holding potential close to 0 mV (average ± SD: 3.9 mV ± 4.1 mV).

## Receptive field estimation

Receptive fields of recorded ganglion cells were estimated by reverse correlation under stimulation with spatiotemporal white noise in a checkerboard layout. Stimulus pixels were 60 or 75 µm on a side, and the intensities of each pixel were drawn randomly and independently from a binary distribution (100% contrast) at either 50 or 30 Hz. Spike-triggered averages were calculated from the 700-ms stimulus segments preceding each spike and separated into spatial and temporal components by singular value decomposition (*Wolfe and Palmer, 1998*). We represented spatial receptive fields by elliptic outlines, obtained from the 1.5-sigma contours of fitted Gaussian profiles, and calculated receptive field diameter as the circle diameter with equal area to the ellipse. Receptive field distances for pairs of cells were determined as the distance between the center points of the fitted Gaussians. Cells with noisy spike-triggered averages were excluded from the receptive field analysis in *Figure 2*.

## Spike-timing correlations

To assess correlations in spike timing for pairs of cells (*Figure 2D,G*), we computed a histogram (bin size 1 ms) of the spiking probability of one ganglion cell, conditioned on the spiking of another ganglion cell, obtained from the spike trains under spatiotemporal white noise. The correlations were quantified by the peak in this histogram, detected in the range from −40 to +40 ms, with the baseline subtracted. The baseline was determined as the average value of the histogram in the ranges from −100 to −20 ms and +20 to +100 ms.

## Saccade stimulus

To mimic the sequence of brief fixations and global image shifts resulting from saccades, we used a spatial square-wave grating of 270 μm spatial period (240 or 300 μm in some experiments) that was repeatedly shifted across the retina for a recording time of typically around 15 min. The grating was presented at 60% Michelson contrast and was held fixed for 800 ms at one of four linearly spaced spatial phases before being shifted during 100 ms to stop again at one of the four spatial phases to start another 800-ms fixation. For each shift, the spatial phases before and afterwards are termed 'starting position' and 'target position', respectively. The sequence of fixation positions was chosen randomly. During each shift, the grating covered a distance of 1.5 to 2.5 spatial periods, depending on the exact difference between starting and target position. For transitions between contrast-reversed gratings, either 2.5 or 1.5 spatial periods could be applied, and we here chose for concreteness to use the former for the two transitions from position 1 to 3 and from 2 to 4 and the latter for the two opposite transitions. The range of transition distances corresponds to shifts of around 13–22° of visual angle, in the range of saccade amplitudes observed in mice (*Sakatani and Isa, 2007*; *Keller et al., 2012*; *Wang et al., 2015*). For the special case where the saccade duration was reduced to 33 ms, the shift was reduced to the range of 0.5 to 1.5 spatial periods to minimize aliasing by the monitor refresh rate. Firing rate profiles were obtained for each transition as peristimulus time histograms (PSTHs) with a bin size of 10 ms. Post-transition peak latency was determined from the PSTH by identifying the bin with the maximum firing rate, fitting a second-order polynomial through this and the two neighboring data points and determining the peak time from the fit.

Variations of the standard stimulus included changes in contrast, in the spatial period of the grating, in the duration of the saccade-like transition, and in the number of fixation positions, as indicated in the text and in the corresponding figures. We also tested the contribution of the motion during the transition by randomly masking the 100-ms transition with uniform gray illumination at mean intensity ('blink-like' transitions; *Figure 3A*). For testing responses to a drift-reset motion, the grating was shifted from its starting position over 266 ms for a distance of half a grating period. This was immediately followed by a rapid shift over 67 ms in the opposite direction of 0.25, 0.5, 0.75, or 1.0 grating periods to reach the target position (*Figure 5B*). The sequence of fixation positions was again chosen randomly.

To test the influence of the spatial extent of the stimulus, we performed experiments during which the saccade stimulus was restricted to a small patch of 480 μm x 480 μm on the retina, surrounded by homogeneous gray illumination at mean intensity (*Figure 4A*). The patch was successively moved to a total of 31 different locations to approximately cover the region spanned by the electrodes. The size of the stimulus patch was chosen to be somewhat larger than typical receptive field centers of IRS cells (see *Figure 2E*). For comparison with responses to the standard stimulus, we included cells only when a stimulus patch completely covered the receptive field center.

To explore responses to saccade-like shifts of natural images, we used photographs from the van Hateren natural image database (*van Hateren and van der Schaaf, 1998*). The images were normalized to the same mean intensity as the gratings and to a contrast of 50% (defined as the standard deviation of the pixel intensity relative to the mean). They were presented with a pixel size of 15 μm on a side and shifted between four fixation positions in the same way as the gratings, translating the image between 600 and 900 μm on the retina. For shifts between equal starting and target position, the image was shifted forth and back between the fixation position and the position in the center of the four fixation positions (*Figure 4C*). Each transition occurred 5–10 times during the recording. To compare saccade-like with blink-like transitions, individual transitions were masked at random by blurring the image during the 100 ms transition period with a Gaussian filter (σ = 2250 μm).

In order to relate the activation of the receptive field by the natural images to the size of the firing rate peaks (inset of *Figure 4C*), we computed for each cell the integrated darkening at transition onset and fixation onset. Five of 17 recorded cells were excluded here because no receptive field had been recovered from the spatiotemporal white-noise analysis. We restricted the analysis to transitions with equal starting and target position and to transitions masked by the blurred image in order to be independent of the complex stimulus dynamics during a motion transition. To compute the activation at, for example, fixation onset, we subtracted the blurred image from the actual image to obtain a measure of contrast for each pixel. To take the receptive field profile into account, we used a 2D Gaussian fit of the spatial receptive field and multiplied each pixel contrast value by the value of the Gaussian profile at the corresponding location. To focus on the negative contrast (since the IRS cells are Off-type cells), we set all positive pixel values to zero, then sign-inverted all pixel values and summed them up over space. We then computed, for each cell and image, the correlation coefficient R between the four activation values (corresponding to the four fixation positions) and the peak size of the second peak in the PSTHs. Finally, we tested whether the collection of these R values significantly deviated from a distribution around zero by a Wilcoxon signed-rank test. The analysis for transition onset proceeded in an analogous fashion, except that we here subtracted the actual image from the blurred image (instead of the other way around) and that the size of the first firing rate peak was used for the correlation.

In the patch-clamp recordings, in order to increase data collection efficiency, fixation periods were shortened to 500 ms, and saccade stimuli were typically presented with only two fixation positions, corresponding to contrast-reversed versions of the same grating. The transitions were masked by a gray screen at mean illumination, and the sequence of fixations was organized so that image recurrences and changes alternated. Furthermore, the grating period in these experiments was set to 200 µm, and the stimulus region was restricted to a 400 µm x 400 µm square, centered on the recorded cell's soma, to reduce cell-by-cell variability in the surround-evoked inhibition.

## Detection of IRS cells

To define a criterion for including recorded cells into the analysis of the IRS response, we evaluated peaks in the firing rate profile after fixation onset, using a window from 50 to 200 ms after fixation onset. First, we required that IRS cells had at least an average peak firing rate of 50 Hz in the PSTHs for the four stimulus transitions with equal starting and target position. This criterion aimed at excluding cells that did not respond to image recurrence and at avoiding false positives that showed spurious firing rate peaks because of noise in the PSTHs. 963 cells out of a total of 2357 recorded cells passed this criterion. We then compared the maxima of the derivatives of the PSTHs between transitions with image recurrence and transitions with maximally different starting and target position. We chose this measure because it is independent of the background activity that the peak my ride on. The derivatives were determined by computing the differences for successive 10-ms time bins in the PSTH. Concretely, we compared for each target image $i = 1, \ldots, 4$ the maximal derivative $D_i^{(\mathrm{rec})}$ under image recurrence (that is when the starting image also equaled *i*) to the maximal derivative $D_i^{(\mathrm{change})}$ when the starting image was the contrast-reversed grating. We then computed a Recurrence Sensitivity Index (*RSI*) as $RSI = \frac{1}{4} \cdot \sum_{i=1}^{4} \left( D_i^{(\mathrm{rec})} - D_i^{(\mathrm{change})} \right) / \left( D_i^{(\mathrm{rec})} + D_i^{(\mathrm{change})} \right)$. Cells with $RSI \geq 0.5$ were considered as IRS cells. This yielded altogether a total of 209 detected IRS cells.

## Model simulation

We modelled the responses of an IRS ganglion cell to saccade-like shifts of a grating according to the neural circuit displayed in *Figure 8A*, representing cells by temporal filters and synaptic transmission by half-wave rectification as shown in *Figure 8B*. The receptive field of the ganglion cell was composed of four non-overlapping spatial subunits that together covered one spatial period of the grating. Each subunit contains an Off-type bipolar cell, which provides excitatory input to the ganglion cell, and an On-type bipolar cell, which is the source of inhibitory input to the ganglion cell via an intermediate On-type amacrine cell in the subunit. The ganglion cell summed over the excitatory and inhibitory inputs from all subunits, and its half-wave rectified activation was interpreted as the firing rate (in arbitrary units). As a critical circuit element, each subunit also contained serial inhibition, mediated by an Off-type amacrine cell, which received excitatory input from the Off-type

bipolar cell and inhibited the On-type amacrine cell. By assuming that the On-type amacrine cell, which suppresses the ganglion cell activity, is glycinergic and that the Off-type amacrine cell, whose influence on the ganglion cell is indirect, is GABAergic, this circuit is consistent with the observed response-increasing action of the gabazine blocker strychnine and with the indirect, response-suppressive action of the GABA_A blocker gabazine (cf. *Figure 7*).

Note that the alignment of individual bipolar cells with quarter-periods of the grating was merely done for simplifying the circuit and is not relevant for its operation. For a better match with physiology, each individual model bipolar or amacrine cell should be thought of as a collection of several, smaller cells, which all receive approximately identical inputs because they are stimulated by the same region of the grating. Furthermore, for simplicity, we restricted the On-type inhibition to act directly on the ganglion cell. Yet, letting this inhibition instead act on the synaptic terminals of the Off-type bipolar cells leads to a functionally nearly identical model. We implemented this by subtracting the inhibitory signal of each On-type amacrine cell from the four Off-type bipolar cell inputs to the ganglion cell before the synaptic rectification instead of afterwards. To keep the net strength of this inhibition fixed, we multiplied the activity of the On-type amacrine cell by 0.5, since the grating activates two Off-type bipolar cells simultaneously. The results from this 'presynaptic-inhibition' model are compared to the standard, 'postsynaptic' model in *Figure 8—figure supplement 2A*.

Finally, note that, in principle, the GABAergic inhibition could also – in order to have an indirect, sign-inverting effect – act on the On-type bipolar cell, which provides the input to the On-type inhibition. However, this would require that the On-type bipolar cell itself be slow enough so that its activity would control the level of inhibition well beyond the stimulation of the On-type bipolar cell during the transition. As this 'long memory' of bipolar cells appears less likely, we here focused on the described direct interaction of glycinergic and GABergic inhibition for exploring their opposing effects.

Visual stimulation, neuronal processing, and synaptic transmission were implemented as follows: The grating stimulus was represented by contrast values of ±1 for bright and dark stripes, respectively. The activation of the bipolar cells was modeled by averaging the contrast signal over the spatial region of the subunit and then applying a biphasic temporal filter to this spatially averaged contrast signal. The temporal filters of the On-type and Off-type bipolar cells were identical except for their sign and had the form (here shown for the Off-type cell)

$$f_{BC}(t) = \frac{1}{Z_1}\left[\exp\left(-\frac{t^2}{T_1^2}\right) - \exp\left(-\frac{t^2}{(2T_1)^2}\right)\right] - \frac{1}{Z_2}\left[\exp\left(-\frac{t^2}{T_2^2}\right) - \exp\left(-\frac{t^2}{(2T_2)^2}\right)\right]$$

for $t>0$ and $f_{BC}(t) = 0$ otherwise, with time constants $T_1 = 40\,\mathrm{ms}$ for the excitatory lobe of the filter and $T_2 = 60\,\mathrm{ms}$ for the antagonistic lobe, and with normalization factors $Z_1$ and $Z_2$, which are used to set the time integral of the filter to zero (to enforce transient activation upon contrast steps) and to normalize it to unit Euclidean norm for a discrete sampling at 0.1 ms resolution.

Synaptic signal transmission was modelled by half-wave rectifying the activation of the presynaptic cell and providing the resulting signal as an additive input to the postsynaptic cell. To model the inhibitory effect of the amacrine cells, their output signals were multiplied by $-1$. The excitatory and inhibitory inputs to the On-type amacrine cell were adjusted by gain factors of 0.05 and 0.2, respectively, to avoid overwhelmingly strong inhibition to the ganglion cell and allow effective suppression of this inhibition by the Off-type amacrine cell. The On-type amacrine cell furthermore contributed temporal low-pass filtering to its input, consistent with the slower time course of glycinergic inhibition in the retina as compared to GABA_A-mediated inhibition (*Moore-Dotson et al., 2015*). The temporal filter was given by

$$f_{AC}(t) = \frac{1}{Z_{AC}}\left[\exp\left(-\frac{t^2}{T_{AC,decay}^2}\right) - \exp\left(-\frac{t^2}{T_{AC,rise}^2}\right)\right]$$

for $t > 0$ and $f_{AC}(t) = 0$ otherwise, with time constants $T_{AC,rise} = 30\,\mathrm{ms}$ and $T_{AC,decay} = 200\,\mathrm{ms}$ and a factor $Z_{AC}$ used for normalization to unit Euclidean norm at 0.1 ms sampling. In addition, this amacrine cell included a tonic activation level, even when it received no input, consistent with the strong tonic inhibition observed in transient Off-alpha ganglion cells (*Zaghloul et al., 2003*; *van Wyk et al.,*

*2009*). The tonic activation was modeled by adding a constant $A_0$ to the cell's filtered input, which we here either set to $A_0 = 2$ (standard case) or $A_0 = 16$ (case of increased tonic inhibition).

Together, the model comprised the following free parameters: the connection strength from the On-type bipolar cell to the On-type amacrine cell, the connection strength from the Off-type amacrine cell to the On-type amacrine cell, the level of tonic activation of the On-type amacrine cell, two time-scale parameters for the bipolar cell filter, and two time-scale parameters for the low-pass filtering by the On-type amacrine cell. Note, though, that we did not perform a systematic optimization of the free parameters to fit the data because the model rather aims at a qualitative proof-of-principle explanation of IRS responses. The code for simulation is available at *Gollisch, 2017* (with a copy archived at https://github.com/elifesciences-publications/IRScellSimulation).

## Acknowledgements

We thank J Demb, M Helmstaedter, and T Moser for helpful comments on a previous version of the manuscript.

## Additional information

### Funding

| Funder | Grant reference number | Author |
|---|---|---|
| Deutsche Forschungsgemeinschaft | Collaborative Research Center 889, project C1 | Tim Gollisch |
| Deutsche Forschungsgemeinschaft | GO 1408/2-1 | Tim Gollisch |

The funders had no role in study design, data collection and interpretation, or the decision to submit the work for publication.

### Author contributions

VK, Conceptualization, Software, Formal analysis, Investigation, Visualization, Methodology, Writing—original draft, Writing—review and editing; MW, Software, Formal analysis, Investigation, Visualization, Methodology, Writing—review and editing; TG, Conceptualization, Resources, Software, Formal analysis, Supervision, Funding acquisition, Investigation, Visualization, Methodology, Writing—original draft, Project administration, Writing—review and editing

### Author ORCIDs

Tim Gollisch, http://orcid.org/0000-0003-3998-533X

### Ethics

Animal experimentation: All experimental procedures were performed in accordance with national and institutional guidelines and approved by the institutional animal care committee of the University Medical Center Goettingen (protocol number T11/35).

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
