## [Decision Letter]

Thank you for submitting your article "Sensitivity to image recurrence across eye-movement-like image transitions through local serial inhibition in the retina" for consideration by *eLife*. Your article has been reviewed by three peer reviewers, one of whom, Alexander Borst (Reviewer #1), is a member of our Board of Reviewing Editors, and the evaluation has been overseen by David Van Essen as the Senior Editor. The following individuals involved in review of your submission have agreed to reveal their identity: Kevin L Briggman (Reviewer #3).

The reviewers have discussed the reviews with one another and the Reviewing Editor has drafted this decision to help you prepare a revised submission.

In this manuscript, Krishnamoorthy and colleagues examine RGC responses to eye-movement like visual stimuli in the mouse retina. They find that 5-10% of RGCs are "image-recurrence sensitive," meaning that their spiking is generally suppressed after a transition from one spatial pattern to the next, unless the same pattern reappears. IRS cells correspond to transient Off α RGCs. Through a series of clever manipulations of a shifting grating stimulus, pharmacology, intracellular recordings and computational modeling, they convincingly show that the IRS response is mediated by a serial inhibition circuit in the inner retina. The authors argue that this type of feature sensitivity may be useful in guiding micro saccades to correct for fixational drift. The mechanistic explanation of this unexpected IRS response is very solid, and it stands as an unusually convincing example of attributing some feature of retinal computation to a serial inhibition circuit. This is achieved mostly through the authors' clever stimulus design and willingness / ability to use whole cell recordings from identified OFF α cells. The effort to firmly identify these IRS cells as the transient OFF α RGCs is very nice.

Major points to address in revision:

1) Function of IRS cells for natural eye movements in mice: do mice show saccadic eye movements, with saccadic intrusions, that would help to stabilize and fixate the image, as the authors speculate in the discussion? The references given in this context by the authors appear to mainly address humans. What about saccades along contours, as in the Discussion? Relatedly, the IRS responses seen using natural images are much less consistent than those elicited by gratings stimuli. From a population perspective this suggests that for a given transition, some subset of cells will show IRS-like responses and others will not. Some discussion of encoding by this heterogeneous population would be useful to include.

2) IRS brain target region: Given the fact that the ganglion cell type is identified, the authors could discuss to which target region in the mouse brain this ganglion cell is sending its axon to.

3) Sources of response variability: From the data it is difficult to get a sense of the sources of variability in the responses. The proposed mechanism behind IRS seems to depend on the appropriate amount of negative contrast within the RF. Similarly, if the variable transition response (e.g Figure 1) is an offset response, then it would rely on there being sufficient positive contrast in the starting position. Can you explain any of the variability in these responses based on the portion of the grating that is within each cell's RF? This sort of analysis would be especially useful in the context of the natural image experiments (Figure 4), which show much less robust IRS responses. Can you use the computational model to predict which natural images will or will not show an IRS response?

4) Illustrating the model: The computational model is a bit hard to follow and ultimately is not used as much as it could have been. It would help the reader to understand the underlying mechanism to show the responses of various model components during some of these stimuli (Figure 8). For example, you could highlight two regions of the stimulus in Figure 8, one that changes and one that doesn't, and illustrate the responses of some of the key model components during and after the transition. Adding a panel between 8B and 8C showing the time series responses of each of the cell types in the model in responses to different transition types would I think make it far easier to grasp. Also, adding a cartoon of the actual anatomy to complement 8A would be a nice addition.

5) Model parameter sensitivity: Some statement about parameter sensitivity in the model would be appreciated. Also a concise statement of the total number of free parameters in the model would help.

6) Evidence for AII: The evidence that the slow ON amacrine cell in this circuit is the AII amacrine is largely circumstantial and based on past work under different conditions. I would suggest softening the focus on this specific amacrine cell and instead highlight the proposed general circuit organization.

---

## [Author Response]

*Major points to address in revision:*

*1) Function of IRS cells for natural eye movements in mice: do mice show saccadic eye movements, with saccadic intrusions, that would help to stabilize and fixate the image, as the authors speculate in the discussion? The references given in this context by the authors appear to mainly address humans. What about saccades along contours, as in the Discussion? Relatedly, the IRS responses seen using natural images are much less consistent than those elicited by gratings stimuli. From a population perspective this suggests that for a given transition, some subset of cells will show IRS-like responses and others will not. Some discussion of encoding by this heterogeneous population would be useful to include.*

Indeed, most information on complex gaze shifting patterns, such as saccadic intrusions, and their dependence on visual stimuli come from humans, and unfortunately, data for mice are still scarce, despite the fact that the existence of saccadic eye movements is now well documented. Complications for obtaining such data come from the fact that eye movements in body-restrained mice are considerably reduced (e.g. Wang et al., J Neurosci 2015) and that both head and eye movements can contribute to gaze shifts. We now comment on the lack of mouse gaze shift data in the Discussion under “Consequences for neural coding”.

Regarding the variability of the IRS responses under shifted natural images, this likely comes from variations in the visual patterns that stimulate the receptive field (see also the response below to point 3). From the perspective of encoding image recurrences, it seems sufficient if a subset of the IRS cells displays a robust IRS response after an image recurrence to provide a robust IRS population response -- likely those IRS cells that have sufficient structure within their receptive fields for this particular fixation. For different images or fixations, different subsets of IRS cells would then contribute to this population response. We now explain this potential population code for image recurrences in the context of the responses under natural images (Figure 4).

*2) IRS brain target region: Given the fact that the ganglion cell type is identified, the authors could discuss to which target region in the mouse brain this ganglion cell is sending its axon to.*

Transient Off-α cells are known to project to both LGN and superior colliculus (Huberman et al., Neuron 2008; Hong et al., J Comp Neurol 2011). We have included a corresponding section in the Discussion under “Cell type identity of IRS cells”, where we now also draw the connection to the known role of the superior colliculus in controlling saccades.

*3) Sources of response variability: From the data it is difficult to get a sense of the sources of variability in the responses. The proposed mechanism behind IRS seems to depend on the appropriate amount of negative contrast within the RF. Similarly, if the variable transition response (e.g Figure 1) is an offset response, then it would rely on there being sufficient positive contrast in the starting position. Can you explain any of the variability in these responses based on the portion of the grating that is within each cell's RF? This sort of analysis would be especially useful in the context of the natural image experiments (Figure 4), which show much less robust IRS responses. Can you use the computational model to predict which natural images will or will not show an IRS response?*

Yes, indeed, the strength of the responses, both at transition onset as well as at fixation onset after image recurrence, should depend on the amount of darkening inside the receptive field, and this should be expected to lead to variability between the responses for different images or different fixation positions. As suggested, we now tested this by computing the correlation between the amount of darkening inside the receptive field and the corresponding response. We found systematic positive correlations both for transition onset and fixation onset and report on these findings in Figure 4. This indicates that, as expected, variations of stimulus structure inside the receptive field are an important source of response variability. We did not aim at predicting these variations through our model because this would require substantial additional assumptions and parameters, regarding the size and placement of the actual receptive fields of the circuit’s interneurons. We therefore feel that our data-analytic approach here allows a more direct connection of the variations in stimulus structure and response strength.

*4) Illustrating the model: The computational model is a bit hard to follow and ultimately is not used as much as it could have been. It would help the reader to understand the underlying mechanism to show the responses of various model components during some of these stimuli (Figure 8). For example, you could highlight two regions of the stimulus in Figure 8, one that changes and one that doesn't, and illustrate the responses of some of the key model components during and after the transition. Adding a panel between 8B and 8C showing the time series responses of each of the cell types in the model in responses to different transition types would I think make it far easier to grasp. Also, adding a cartoon of the actual anatomy to complement 8A would be a nice addition.*

As suggested, we have included a panel of the activation of the local interneurons in the model to different local stimulus patterns (now Figure 8). This should indeed make it easier to understand how this model circuit operates. In particular, this shows that locations with net brightening across the transition provide slow, long-lasting inhibition, which suppresses activity after fixation onset. For recurring images, on the other hand, inhibition becomes suppressed after fixation onset at all locations, from darkening either at transition onset or at fixation onset. We have added explanations along these lines in the Results section when discussing the new Figure 8.

Note that, in order to facilitate these explanations and the interpretation of this new figure panel, we have slightly adjusted the model. Besides some of the model parameters, this concerns the target of the On-type inhibition, which previously acted both directly onto the ganglion cell and presynaptically onto the bipolar cells. To simplify the explanations and to focus on the essential model components, the standard model version now uses only direct inhibition onto the ganglion cell. However, we now show in the new Figure 8—figure supplement 2 that presynaptic inhibition can be used just as well without changing the response of the model.

Regarding the circuit anatomy, the schematic circuit diagram in Figure 8 is thought to serve as a cartoon of the anatomy, displaying the putative synaptic connections, complementing the mathematical implementation illustrated in Figure 8. We do not feel that we have sufficient evidence to include further anatomical details about the circuit.

*5) Model parameter sensitivity: Some statement about parameter sensitivity in the model would be appreciated. Also a concise statement of the total number of free parameters in the model would help.*

We have added a supplemental figure (Figure 8—figure supplement 2) which shows model responses when critical model parameters are doubled or halved. This shows that the model does not require fine tuning of parameters to qualitatively reproduce the IRS responses. Furthermore, we added some explanations regarding the model’s free parameters in the methods section. Please note, though, that we did not perform any systematic optimization of the free parameters to fit the experimental data because the model aims at providing a proof-of-principle explanation of the IRS responses rather than a detailed reproduction of the firing rate profiles.

*6) Evidence for AII: The evidence that the slow ON amacrine cell in this circuit is the AII amacrine is largely circumstantial and based on past work under different conditions. I would suggest softening the focus on this specific amacrine cell and instead highlight the proposed general circuit organization.*

Yes, agreed. We have reformulated the corresponding statements to make it clearer that the AII here primarily serves only as an example of glycinergic signaling, which itself might be controlled or modulated by GABAergic inhibition.